

Unbalanced relationship between flood risk perception and flood
preparedness from the perspective of response intention and socio-
economic factors: a case study of Nanjing, China
*Yabo Li [a,b], Peng Wang [b,c,*]*
*[a] Polytechnic Institute, Zhejiang University, Hangzhou 310015, China*
*[b] Faculty of Civil Engineering and Mechanics, Jiangsu University, Zhenjiang 212013,*
*China*
*[c] State Key Laboratory of Pollution Control & Resource Reuse, School of the*
*Environment, Nanjing University, Nanjing 210023, China*

*Corresponding author.

*E-mail address:* upeswp@ujs.edu.cn (P. Wang).





**Abstract**
Although risk perception and flood preparedness were crucial in flood
management, perceived flood risk was not always translated into flood preparedness. It
was essential to investigate the potential association between risk perception and flood
preparedness. This study focused on Nanjing as the research region, designed
questionnaire survey and explored the influence relationship between risk perception
and flood preparedness. Participants showed the medium perception of food risk and
higher flood preparedness. Higher risk perception was observed in individuals with
regular exercising, the elderly, low education level and long living time. Higher flood
preparedness occurred among groups of females, the elderly and high education level.
Individuals relied more on threat appraisal to perceive risk, which failed to trigger high
enough coping appraisal. Inadequate risk perception led to a strenuous transform into
flood preparedness with unbalanced relationship. Groups with distinct socio-economic
characteristics exhibited varying preferences to achieve risk perception and flood
preparedness. Females relied more on flood knowledge to perceive flood risk. Path
analysis suggested that threat appraisal transformed into flood preparedness under the
effect of response intention and social-economic features. Groups with high education
level or bad health were more likely to perceive risk and engage in preventive behavior.
These findings could provide critical insights into intervention strategies for enhancing
public flood preparedness in flood management.
**Keywords: Flood risk perception, flood preparedness, response intention,**
**influence path, flood risk mitigation**



## 1. Introduction

Natural disasters caused immense damage and irreversible losses with global climate change (Guo, Wu, & Wei, 2020). Floods still remained the most prevalent and severe disaster worldwide and occupied the dominated composition in a total of 432 disaster events in 2021. Due to rapid urbanization and concentrated population and assets (Deng, Wang, Wu, & Lai, 2022; Dong et al., 2022), urban areas became more susceptive and vulnerable to flood events (P. Wang, Li, & Zhang, 2021). It was predicted that climate change and heavy rainfall were more frequent and intense with high reliability (Rifat & Liu, 2022; Steinhausen et al., 2021), and significantly increased urban flood risk, especially in developing countries (Zhu et al., 2021). Despite the substantial financial investment and mitigation efforts, floods continued to pose a serious threat to human society in the foreseeable future (Thongs, 2019; Ke Zhang et al., 2022). It was imperative to adopt effective flood management for sustainable development.

In response to flood events, it was unadvisable to completely take traditional structural measures (Rasool, Rana, & Ahmad, 2022), such as dikes and dams. Risk perception acted as non-structural measures and received considerable attention in current research (Ahmad & Afzal, 2020). Flood risk perception reflected risk acceptance and revealed feelings, opinions and judgements regarding direct or potential hazards (Rana, Jamshed, Younas, & Bhatti, 2020; Yang, 2019). According to Protection Motivation Theory (PMT), cognitive process determined self-protective motivation (Khani Jeihooni, Bashti, Erfanian, Ostovarfar, & Afzali Hasirini, 2022), and threat



appraisal and coping appraisal were the important components of risk perception
(Roder, Hudson, & Tarolli, 2019). Limited understanding of flood risk perception led
to failures in flood management practices (Ahmad & Afzal, 2020). Successful flood
management highly depended on the implementation of mitigation measures, because
people were both flooding victims and implementors of disaster mitigation policies (Z.
Wang, Wang, Huang, Kang, & Han, 2018; Yin et al., 2021). Flood preparedness acted
as individual protection action and reflected response behaviors during floods,
including preventive and adaptive behavior (Sado-Inamura & Fukushi, 2019).
Subjective expected utility theory assumed that people assessed likelihood and
consequences of alternative choices (Rufat & Botzen, 2022). Individuals would seek or
wait for sufficient information to support the action of responding to flooding (Dootson,
Kuligowski, Greer, Miller, & Tippett, 2022; Rufat & Botzen, 2022). Adequate flood
preparedness ensured that people could adjust their behaviors more rationally and
effectively, making minor changes to mitigate adverse impacts from floods (Valois et
al., 2020).
Flood risk perception was believed to promote flood preparedness, but high
perceptions of risk could not always translate to disaster preparedness (Schlef, Kaboré,
Karambiri, Yang, & Brown, 2018). There was not direct and simple relationship
between risk perception and flood preparedness as expected. Some studies found the
results contradictory to the popular opinion that high perception of flood risk caused
high flood preparedness (Rasool et al., 2022) and suggested the weak relationship
between risk perception and flood preparedness (Valois et al., 2020), even without



direct link (Ao et al., 2020; Wachinger, Keilholz, & O'Brian, 2018). High risk
perception even motivated people to avoid or ignore willfully specific actions under
uncertain circumstances (Wachinger, Renn, Begg, & Kuhlicke, 2013). There was no
consensus on how risk perception affected and predicted preparedness behavior (Huang
& Lubell, 2022; Taylor, Dessai, & Bruine de Bruin, 2014). Connection between risk
perception and preparedness appeared more strenuous in practice (Valois et al., 2020),
due to the ignorance of the existence of unknown intermediary (Ao et al., 2020; Yong
& Lemyre, 2019). Theory of Planned Behavior anticipated how people behaved in
specific situation and connected behavior with individual control, with intention being
the predictor of behavior (Ghanian et al., 2020; Kurata, Prasetyo, Ong, Nadlifatin, &
Chuenyindee, 2022). For individual cognitive decision-making, intention served as the
intermediate link between perception and behavior (Soetanto, Mullins, & Achour,
2017), and ample social-scientific evidence supported the positive relationship between
risk perception and intention to respond, not actual behaviors (Harlan, Sarango, Mack,
& Stephens, 2019; van Valkengoed & Steg, 2019).
Individuals with different backgrounds got involved in flood management,
perceived flood risk in various ways (Rasool et al., 2022) and developed personal
intention to follow risk response (Kurata et al., 2022). Socio-economic features were
the most controversial driving factors of risk perception (Shah et al., 2020) and flood
preparedness (Ao et al., 2020), with relevant studies reporting mixed and inconsistent
results (Rufat & Botzen, 2022). Socio-economic features determined the social group
to which people belonged and affected individual perception and action towards


hazards (Harlan et al., 2019). But most studies only estimated simple correlations and
included socio-economic factors as control variables in regression analysis (Rufat &
Botzen, 2022). Moreover, most research mainly focused on influencing factors of risk
perception and flood preparedness (Ao et al., 2020; Sun & Sun, 2019; Ullah, Saqib,
Ahmad, & Fadlallah, 2020). The few attached importance to influence path between
flood risk perception and flood preparedness (Wachinger et al., 2018). Existing research
has extensively examined risk perception and flood preparedness in developed nations,
but the potential linkages between flood risk perception and disaster preparedness was
under-explored, particularly in developing countries (Scaini, Stritih, Brouillet, & Scaini,
2021; Keshun Zhang, Parks-Stamm, Ji, & Wang, 2021). Effective policies for flood
management could benefit from a more integrated intervention framework that
connected risk perception with flood preparedness.

Despite the continuous flood protection efforts, Nanjing has experienced

increasingly severe flood damage in recent years. This study examined flood risk
perception in Nanjing, and investigated transformation relationship between risk
perception and flood preparedness from the perspective of response intention and socio-
economic factors. This study aimed to: (1) identify the distribution characteristics of
risk perception and flood preparedness; (2) analyze the influence effect of different
factors combined with social-economic feature; (3) reveal the influence path between
risk perception and flood preparedness. **Fig. 1** illustrated the comprehensive framework
of the study.


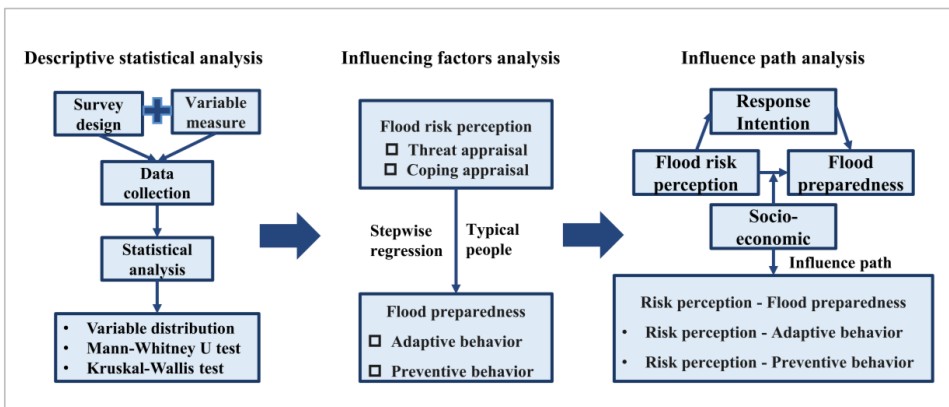


**Fig. 1. Overall framework of this study.**


**2. Material and methods**
**2.1 Study region**

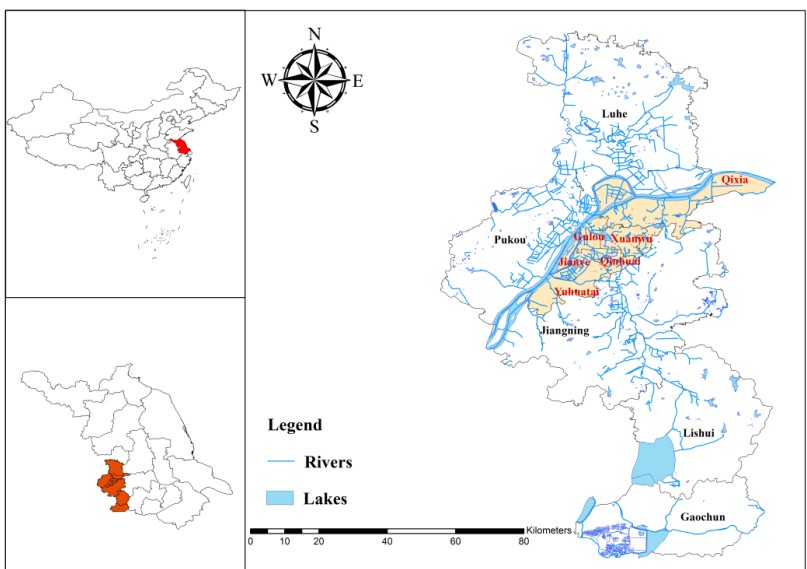


**Fig. 2. Study area**
Nanjing was located in the middle and lower reaches of Yangtze River in eastern
China, with a municipal area of 6587.02 km$^2$. The city belonged to a typical subtropical
and monsoon climate region, characterized by distinct seasonal changes and abundant





rainfall. Nanjing had jurisdiction over 11 urban district, 95 streets and 6 towns by 2021.
As one of national key flood control cities, Nanjing confronted with the conflict of rapid
urbanization and increasing floods (H. Zhang et al., 2021). Nanjing was estimated to
exhibit higher flood risk across various flood return periods (P. Wang, Li, Yu, & Zhang,
2021), especially in the central urban districts surrounding the Yangtze River (Li et al.,
2022). Consequently, this study focused on six districts (**Fig. 2**) of urban center to
explore the relationship between flood risk perception and flood preparedness for
fostering flood resilient cities.
**2.2 Survey design and variables measure**

This study developed a semi-structured questionnaire through Likert scale to

investigate flood risk perception in Nanjing. The survey primarily consisted of four
parts: (1) Socio-economic condition; (2) Flood risk perception; (3) Flood preparedness;
(4) Response intention. Comprehensive and detailed description of the questionnaire
was provided in **Supplementary material**. The first section collected information
about participants' socio-economic condition, including gender, age, district, education
background, living time, physical condition, exercise situation and life style
(particularly bad habits, such as smoking). Based on PMT, the second part measured
flood risk perception by examining both threat appraisal and coping appraisal.

Flood preparedness encompassed both adaptive and preventive behaviors in the

third section. Adaptive behavior involved a series of measures aimed at mitigating and
adapting to floods, while preventive behavior focused on actions taken to prevent and
reduce negative effects during flood events. The fourth section included a survey on


response intention and consisted of factors influencing flood risk perception and
preparedness. Additionally, flood risk knowledge referred to the level of grasping flood
related knowledge among local residents, while flood risk worry evaluated individuals'
fear and concern about floods. Furthermore, flood experience reflected the frequency
of exposure to flood disasters. Government trust revealed the degree of confidence in
government flood management, while flood disaster education measured the diversity
of education resources available for residents regarding floods. **Table 1** presented the
collected indicators and variables from the questionnaire survey.
**Table 1**
Indicator and variable measurement.

| Indicator | Variable | Range |
|---|---|---|
| Flood risk perception | Threat appraisal | (1,5) |
| | Coping appraisal | (1,5) |
| Flood preparedness | Adaptive behavior | (1,5) |
| | Preventive behavior | (1,5) |
| | Flood risk knowledge | (1,5) |
| | Government trust | (1,5) |
| Response intention | Flood risk worry | (0,1) |
| | Flood experience | (0,1) |
| | Flood disaster education | (0,1) |
| | Gender | (1,2) |
| | Age | (1,7) |
| | District | (1,6) |
| Socio-economic factors | Education level | (1,5) |
| | Living time | (1,5) |
| | Health condition | (1,5) |


| Life style | (0,1) |
|---|---|
| Exercise situation | (0,1) |

**2.3 Data collection**
This study conducted a survey in six districts of Nanjing: Gulou, Xuanwu, Jianye,
Qinhuai, Qixia and Yuhuatai district respectively. We implemented the preliminary test
of online questionnaire before officially issue to minimize the participants'
misunderstanding and confusion. Survey results were collected from interviewees and
the questionnaire was adjusted based on online feedback to reduce bias. We then
performed face-to-face questionnaire surveys in densely populated streets of Nanjing
from April 24, 2021 to April 30, 2021.
Interviewers received excellent survey skills training before formal interview, and
were organized into six groups with at least two members in each group. A group leader
was appointed to distribute and collect questionnaires, supervise and record the entire
process, and ensure the rationality and effectiveness of data acquisition. Each
interviewer introduced and emphasized the objectives of this questionnaire survey at
the beginning. Strictly following the principle of voluntary participation and
confidentiality, respondents were given enough time to review questionnaire content
adequately, and had the option to withdraw from survey at any time. Complete
questionnaire comprised 52 questions and required 15–20 minutes for completion
approximately. To encourage and appreciate for participation, interviewers presented
self-made gifts to respondents after finish. Eventually, this study distributed 844
questionnaires and obtained 737 valid questionnaires after excluding 107 invalid ones
with an effective rate of 87.32%.


**2.4 Statistical analysis**

By exporting the collected data to SPSS software, this study calculated each indicator by averaging the corresponding variables, and conducted descriptive analysis to reveal the distribution features of different indicators and variables. Mann-Whitney U and Kruskal-Wallis statistical tests were nonparametric statistical test used to compare the value of variables between two and several independent groups respectively. We employed these tests to compare the average score of indicators and variables and explore whether there were the statistically significant differences in distribution. Correlation analysis were used to examine the influence factors of flood risk perception and flood preparedness. Furthermore, this study implemented the stepwise regression approach to reveal the impact of different factors on risk perception and flood preparedness. Finally, we performed Model 5 in PROCESS macro program of SPSS to capture the influence path between flood risk perception and flood preparedness. All statistical analyses were conducted at a significance level of 0.05.

**3. Results**

**3.1 Descriptive statistical analysis**

Cronbach's α (0894) and KMO value (0.891) both exceeded 0.7 in this questionnaire and illustrated the high reliability and validity. **Table 2** presented the descriptive analysis about basic information of participants. Among 739 respondents, there was a gender distribution with 43.8% males and 56.2% females. Most people were aged from 18 to 25 years (27.5%), followed by 31-40 years (20.8%), 41-50 years





(14.5%), 26-30 years (12.5%), over 60years (11.9%), 51-60 years (11.4%) and below
18 years (1.4%). The majority of participants came from Jianye district (26.2%),
followed by Qixia (23.2%), Gulou (21.8%), Yuhuatai (11.7%), Xuanwu (10.2%), and
Qinhuai District (6.9%). Education level was mostly undergraduate (45.6%), middle
school (16.3%), high school (19.7%), postgraduate and above (11.5%) and elementary
school (6.9%). Regarding their residence duration, most participants lived in Nanjing
for above 10 years (51.4%), 1-3 years (17.0%), 3-5 years (11.9%), 5-10 years (11.9%),
and below 1 years (7.7%). More than half of respondents stayed in excellent health
(49.5%), better (34.7%) and general health (13.6%), while few people reported very
poor (0.4%) and poor (1.8%) health. The majority didn't smoke (81.1%) and 18.9%
showed the habit of smoking. Over half often engaged in regular exercising (61.2%)
and 38.8% lacked adequate exercise.
**Table 2**
Profile of socio-economic feature in respondents.

| Characteristic | Description | Frequency | Rate |
|---|---|---|---|
| Gender | Male | 323 | 43.8 |
| | Female | 414 | 56.2 |
| Age | ≤18 years | 10 | 1.4 |
| | 18-25 years | 203 | 27.5 |
| | 26-30 years | 92 | 12.5 |
| | 31-40 years | 153 | 20.8 |
| | 41-50 years | 107 | 14.5 |
| | 51-60 years | 84 | 11.4 |
| | ≥60 years | 88 | 11.9 |




| | Gulou | 161 | 21.8 |
|---|---|---|---|
| District | Jianye | 193 | 26.2 |
| | Qixia | 171 | 23.2 |
| | Qinhuai | 51 | 6.9 |
| | Xuanwu | 75 | 10.2 |
| | Yuhuatai | 86 | 11.7 |
| | Elementary school | 51 | 6.9 |
| | Middle school | 120 | 16.3 |
| Education level | High school | 145 | 19.7 |
| | Undergraduate | 336 | 45.6 |
| | Postgraduate and above | 85 | 11.5 |
| | Less than 1 years | 57 | 7.7 |
| | 1-3 years | 125 | 17.0 |
| Living time | 3-5 years | 88 | 11.9 |
| | 5-10 years | 88 | 11.9 |
| | More than 10 years | 379 | 51.4 |
| | Very poor | 3 | 0.4 |
| | Poor | 13 | 1.8 |
| Health condition | General | 100 | 13.6 |
| | Better | 256 | 34.7 |
| | Excellent | 365 | 49.5 |
| Life style | Smoking | 139 | 18.9 |
| | Not smoking | 598 | 81.1 |
| Exercise situation | Regularly exercising | 451 | 61.2 |
| | Not exercising | 286 | 38.8 |

**Table 3** showed the score of each variable and indicator. Flood risk perception kept
at a medium level with the average score of 3.57. Residents exhibited a high level of
threat appraisal and a medium level of coping appraisal. The average level of flood





preparedness was relatively high (4.05), and local participants demonstrated the high
level of adaptive behavior (4.25) and medium level of preventive behavior (3.85).
Furthermore, a medium level of flood risk knowledge and government trust was
observed among respondents (2.73 and 2.94). There was also a low level of flood
experience and flood disaster education (0.45 and 0.46). Flood risk worry showed a
medium level (0.50), while participants had a relatively low level in response intention

(2.73).

**Table 3**
Descriptive statistics of each indicator and variable.

| Name | Min | Max | Mean | Standard Deviation |
|---|---|---|---|---|
| Flood risk perception | 1 | 5 | 3.57 | 0.68 |
| • Threat appraisal | 1 | 5 | 4.10 | 0.61 |
| • Coping appraisal | 1 | 5 | 3.03 | 1.07 |
| Flood preparedness | 1 | 5 | 4.05 | 0.76 |
| • Adaptive behavior | 1 | 5 | 4.25 | 0.79 |
| • Preventive behavior | 1 | 5 | 3.85 | 0.87 |
| Response intention | 1 | 5 | 2.87 | 0.79 |
| • Flood risk knowledge | 1 | 5 | 2.73 | 1.25 |
| • Government trust | 1 | 5 | 2.94 | 0.50 |
| • Flood risk worry | 0 | 1 | 0.50 | 1.15 |
| • Flood experience | 0 | 1 | 0.45 | 0.25 |
| • Flood disaster education | 0 | 1 | 0.46 | 0.50 |
| Gender | 1 | 2 | 1.56 | 0.50 |
| Age | 1 | 7 | 4.01 | 1.74 |
| District | 1 | 6 | 2.92 | 1.63 |
| Education level | 1 | 5 | 3.39 | 1.10 |



| Living time | 1 | 5 | 3.82 | 1.41 |
|---|---|---|---|---|
| Health condition | 1 | 5 | 4.31 | 0.80 |
| Life style | 0 | 1 | 0.19 | 0.39 |
| Exercise situation | 0 | 1 | 0.61 | 0.49 |

**3.2 Distribution test**

**Table 4-7** presented significant results from Mann-Whitney U test. In gender category, there were significant differences in adaptive behavior, preventive behavior, flood preparedness, flood risk worry and government trust. Males' mean rank was 340.71, 336.66, 338.06, 343.22 and 392.47, while females demonstrated the mean rank of 391.07, 394.23, 393.14, 389.11 and 350.69 respectively. Women exhibited a higher level in flood preparedness, adaptative and preventive behavior, and flood risk worry, while men had a higher level of government trust. Regularly exercising people had higher levels of threat appraisal and flood risk perception, with average ranks of 389.37 and 385.47, compared to those who did not exercise (336.88 and 343.02). Furthermore, individuals with flood risk worry exhibited higher levels of flood risk perception, flood preparedness and response intention, with mean rank of 387.33, 397.41 and 479.18. Groups with flood experience showed higher levels of flood risk perception and response intention (416.08 and 507.11).

**Table 4**

Mann-Whitney U test in gender.

| Category | Gender | | | |
|---|---|---|---|---|
| | Mean rank | | Z-value | P-value |
| | Male | Female | | |
| Adaptive behavior | 340.71 | 391.07 | -3.22 | 0.00 |





| | | | | |
|---|---|---|---|---|
| Preventive behavior | 336.66 | 394.23 | -3.65 | 0.00 |
| Flood preparedness | 338.06 | 393.14 | -3.49 | 0.00 |
| Flood risk worry | 343.22 | 389.11 | -3.35 | 0.00 |
| Government trust | 392.47 | 350.69 | -2.65 | 0.01 |

**Table 5**
Mann-Whitney U test in exercise situation.

| Category | Exercise situation | | Z-value | P-value |
|---|---|---|---|---|
| | Mean rank | | | |
| | Regularly exercising | Not exercising | | |
| Flood risk perception | 385.47 | 343.02 | -2.64 | 0.01 |
| Threat appraisal | 389.37 | 336.88 | -3.28 | 0.00 |

**Table 6**
Mann-Whitney U test in exercise situation.

| Category | Flood risk worry | | Z-value | P-value |
|---|---|---|---|---|
| | Mean rank | | | |
| | Yes | No | | |
| Flood risk perception | 387.33 | 350.42 | 0.02 | 0.02 |
| Threat appraisal | 398.91 | 338.68 | -3.86 | 0.00 |
| Flood preparedness | 397.41 | 340.20 | 0.00 | 0.00 |
| Adaptive behavior | 386.47 | 351.29 | 0.02 | 0.02 |
| Preventive behavior | 401.09 | 336.47 | 0.00 | 0.00 |
| Response intention | 479.18 | 257.32 | 0.00 | 0.00 |

**Table 7**
Mann-Whitney U test in flood experience.

| Category | Flood experience | | Z-value | P-value |
|---|---|---|---|---|
| | Mean rank | | | |
| | Yes | No | | |

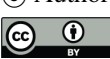


| | | | | |
|---|---|---|---|---|
| Flood risk perception | 416.08 | 330.62 | 0.00 | 0.00 |
| Coping appraisal | 419.43 | 327.88 | 0.00 | 0.00 |
| Response intention | 507.11 | 256.41 | 0.00 | 0.00 |

**Table 8-11** displayed significant results from Kruskal-Wallis statistical test.
Among age groups, individuals aged 31 to 40 showed a higher level in threat appraisal
than those aged 18 to 25. Coping appraisal levels were lower in the 18-25 age group
compared to those aged 51-60 and ≥60 years. Preventive behavior was lower among
people aged 51-60 than those aged 18-25 and 31-40, respectively. People aged 51-60
possessed more flood risk knowledge than those aged 18-25. Government trust was
higher among individuals aged under 18 and 41 to 50 than aged 51 to 60. Flood disaster
education level was higher in the 41-50 age group than in18-25 age group. Flood risk
perception was higher among individuals aged 51-60 and over 60 years than those aged
18-25. Flood preparedness was higher among individuals aged 31 to 40 than those aged
51 to 60 years, while response intention was higher within people aged 51-60 years than
those aged 18-25.
**Table 8**
Kruskal-Wallis test in age.

| Category | Age | | | | | | | Sig.(p) |
|---|---|---|---|---|---|---|---|---|
| | Mean rank | | | | | | | |
| | ≤18 | 18-25 | 26-30 | 31-40 | 41-50 | 51-60 | ≥60 | |
| Threat appraisal | – | 325.28 | – | 389.43 | – | – | – | 0.000 |
| Coping appraisal | – | 324.17 | – | – | – | 447.88 | 410.81 | 0.000 |





| | | | | | | | | |
|---|---|---|---|---|---|---|---|---|
| Preventive behavior | 373.19 | 388.55 | – | 402.44 | – | 298.23 | – | 0.000 |
| Flood risk knowledge | – | 328.55 | – | – | – | 442.48 | – | 0.001 |
| Government trust | 543.8 | – | – | – | 414.45 | 312.82 | – | 0.005 |
| Flood disaster education | – | 397.46 | – | – | 305.3 | – | – | 0.004 |
| Flood risk perception | – | 321.87 | – | – | – | 418.44 | 405.55 | 0.000 |
| Flood preparedness | – | – | – | 414.17 | – | 315.46 | – | 0.009 |
| Response intention | – | 333.9 | – | – | – | 425.55 | – | 0.021 |

In terms of education level (**Table 9**), the average rank of threat appraisal for
postgraduate and above was lower than that of high school and undergraduate. Coping
appraisal for postgraduate was lower than that of middle school, high school and
undergraduate. People with an undergraduate education exhibited a higher mean rank
of preventive behavior than those in middle school. People with middle school and high
school education possessed a higher level of flood risk knowledge than that of
postgraduates and above. There was a higher level of flood disaster education at the
undergraduate level than that of middle and high school. Individuals with postgraduate
and higher levels of education showed a lower level of flood risk perception than those
in middle school. Additionally, individuals with the undergraduate degree demonstrated
a higher level of flood preparedness compared to those in middle school.
**Table 9**





Kruskal-Wallis test in education level.

| Category | Education level | | | | | |
| | Mean rank | | | | | |
| | Elementary school | Middle school | High school | Under-graduate | Postgraduate and above | Sig.(p) |
| --- | --- | --- | --- | --- | --- | --- |
| Threat appraisal | – | – | 383.63 | 382.05 | 296.02 | 0.000 |
| Coping appraisal | – | 399.48 | 399.89 | 366.10 | 291.75 | 0.001 |
| Preventive behavior | – | 330.55 | – | 403.93 | – | 0.001 |
| Flood risk knowledge | – | 393.72 | 398.97 | – | 300.49 | 0.009 |
| Flood disaster education | – | 325.10 | 335.67 | 395.81 | – | 0.003 |
| Flood risk perception | – | 382.68 | 406.71 | – | 298.69 | 0.000 |
| Flood preparedness | – | 330.47 | – | 400.63 | – | 0.004 |

Moreover, people residing for over 10 years had the higher mean rank of coping
appraisal than those living for less than 1 year, 1-3 years and 5-10 years. Living for less
than 1 year brought a low level in coping appraisal than residing for 3-5 years.
Individuals with residence duration of over 10 years grasped more flood risk knowledge
than living time of less than 1 year, 1-3 years and 5-10 years. Mean rank of flood
experience was higher for individuals residing for over 10 years than those living for
less than 1 year, 1-3 years, and 3-5 years. People with over 10 years living time had a





higher level of flood risk perception and response intention than those residing for less
than 1 year and 1-3 years. In **Table 11**, as physical health improved from better to
excellent, there was the increasing trend in the mean rank of threat appraisal and flood
risk perception. People with excellent health exhibited a higher level in preventive
behavior than those with general health. And general and better health conditions had
the lower mean rank of government trust than excellent health.
**Table 10**
Kruskal-Wallis test in living time.

| Category | Living time | | | | | Sig.(p) |
| | Mean rank | | | | | |
| | < 1 years | 1-3 years | 3-5 years | 5-10 years | >10 years | |
|---|---|---|---|---|---|---|
| Coping appraisal | 246.36 | 317.28 | 354.16 | 337.88 | 415.18 | 0.000 |
| Flood risk knowledge | 259.13 | 311.44 | – | 33.33 | 414.90 | 0.000 |
| Flood experience | 326.33 | 330.26 | 329.12 | – | 402.82 | 0.000 |
| Flood risk perception | 275.73 | 318.74 | – | – | 409.12 | 0.000 |
| Response intention | – | 319.23 | – | 322.77 | 406.30 | 0.000 |

**Table 11**
Kruskal-Wallis test in health condition.

| Category | Health condition | |
| | Mean rank | Sig.(p) |
|---|---|---|





|  | Very poor | Poor | General | Better | Excellent |  |
|---|---|---|---|---|---|---|
| Coping appraisal | – | – | – | 329.43 | 400.45 | 0.000 |
| Preventive behavior | – | – | 326.03 | – | 399.35 | 0.001 |
| Government trust | – | 200.35 | 308.91 | – | 392.22 | 0.000 |
| Flood risk perception | – | – | – | 342.38 | 390.58 | 0.009 |

**3.3 Correlation analysis**

In **Fig. 3**, flood risk knowledge was significantly and positively related to coping appraisal and flood risk perception. There was the moderately positive and significant correlation between government trust and flood risk perception. Flood risk worry, flood disaster education and flood experience showed significant and weakly positive relationship with risk perception. Among socio-economic factors, gender exhibited no significant correlation with flood risk perception, and other variables were weakly related to flood risk perception. Government trust was significant and moderately positive correlated with flood preparedness, while flood risk knowledge, flood risk worry, flood disaster education and flood experience showed weakly related to flood preparedness. Only district, education level, living time, life style and exercise situation were unrelated to flood preparedness. Gender, age and health condition were weakly correlated to flood preparedness. Flood risk perception was significantly and positively related to response intention, but flood preparedness showed lower correlation with flood risk perception and intention response.

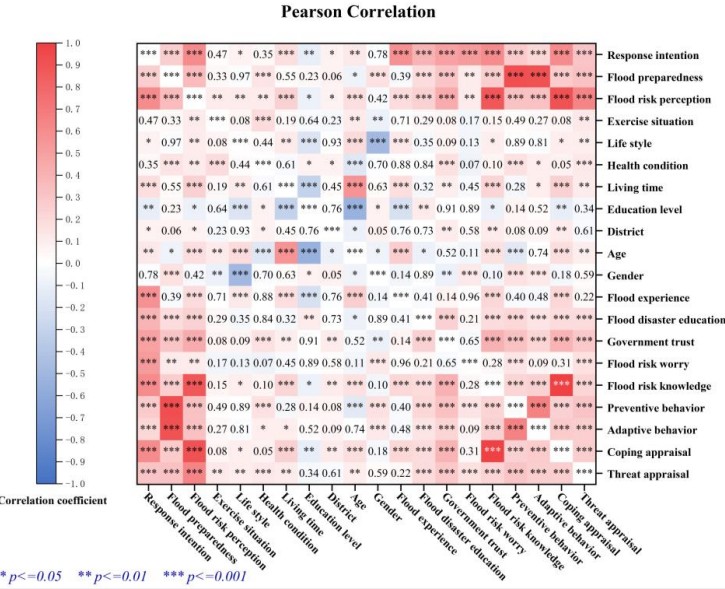

*p<=0.05    **p<=0.01    ***p<=0.001

**Fig. 3. Pearson correlation analysis.**

**3.4. Influencing factors of risk perception**

**Table 12** presented the results of stepwise regression analysis. We selected all variables for regression analysis in model 1, and found that flood risk knowledge showed significant and positive effect while other variables exhibited relatively lower effects. Model 2 demonstrated a high goodness of fit (adjusted $R^2$=0.788) after removing socio-economic variables, and flood risk knowledge also maintained a higher influence (0.827) on flood risk perception. In model 3, the exclusion of flood risk knowledge resulted in a low goodness of fit (adjusted $R^2$=0.246). But government trust, flood experience, flood disaster education and flood risk worry significantly and positively influenced risk perception, indicated by increased regression coefficients. The effect of flood experience on risk perception shifted from insignificant to significant. Although flood risk knowledge significantly promoted risk perception, it



also inhibited and decreased the positive effects of other factors. Faced with insufficient
flood risk knowledge, maintaining trust in government and recalling past flooding
experience were crucial for enhancing flood risk perception.
**Table 12**
Stepwise regression analysis results of flood risk perception.

| Variable | Standardized coefficient | | |
|---|---|---|---|
| | Model 1 | Model 2 | Model 3 |
| Flood risk knowledge | 0.814*** | 0.827*** | - |
| Flood risk worry | 0.074*** | 0.067*** | 0.100** |
| Government trust | 0.093*** | 0.094*** | 0.396*** |
| Flood disaster education | 0.060*** | 0.053*** | 0.146*** |
| Flood experience | -0.010*** | 0.010 | 0.168*** |
| Gender | 0.057** | - | - |
| Age | 0.067** | - | - |
| District | -0.027 | - | - |
| Education level | 0.010 | - | - |
| Living time | 0.010 | - | - |
| Health condition | 0.056** | - | - |
| Life style | 0.057** | - | - |
| Exercise situation | 0.038* | - | - |
| $R^2$ | 0.803 | 0.790 | 0.250 |
| Adjusted $R^2$ | 0.800 | 0.788 | 0.246 |
| F | 227.270 | 549.538 | 61.083 |

*** $P < 0.001$, ** $P < 0.01$, * $P < 0.05$

This study categorized participants based on socio-economic feature to explore the
impact of different factors. **Fig. 4** only listed the significant results of regression
analysis and more detailed information was provided in **Supplementary materials**.





Among males, flood risk knowledge, flood risk worry, government trust and flood
disaster education positively affected flood risk perception, with standardized
coefficients of 0.815, 0.087, 0.105 and 0.062 respectively. In females, flood risk
knowledge, flood risk worry and government trust exhibited significant effects on risk
perception, with standardized coefficients of 0.841, 0.043 and 0.090 respectively. Flood
risk knowledge showed a greater impact among females, while flood risk worry and
government trust had a higher influence in males. Among the elderly, flood risk
knowledge and worry significantly affected flood risk perception with influence
coefficients of 0.828 and 0.128 respectively. Flood risk knowledge, flood risk worry,
government trust and flood disaster education showed significant effects (0.823, 0.059,
0.101 and 0.056) among young and middle-aged individuals. Compared with the non-
elderly, the elderly exhibited a higher influence of flood risk knowledge and worry on
risk perception.
In people with high education level, flood risk knowledge and government trust
significantly and positively affected flood risk perception (0.817 and 0.124). However,
for individuals with low education level, flood risk knowledge showed a higher impact
(0.831), and flood risk worry and flood disaster education significantly influenced risk
perception, with standardized coefficients of 0.109 and 0.093 respectively. For
individuals with a short living time, only flood risk knowledge and government trust
showed significant positive effects (0.734 and 0.187) to flood risk perception. But
among people with long living time, flood risk knowledge had a greater impact on risk
perception (0.829), while government trust exhibited a lower effect (0.064).


Additionally, flood risk worry and disaster education also showed significant effects
(0.051 and 0.083).

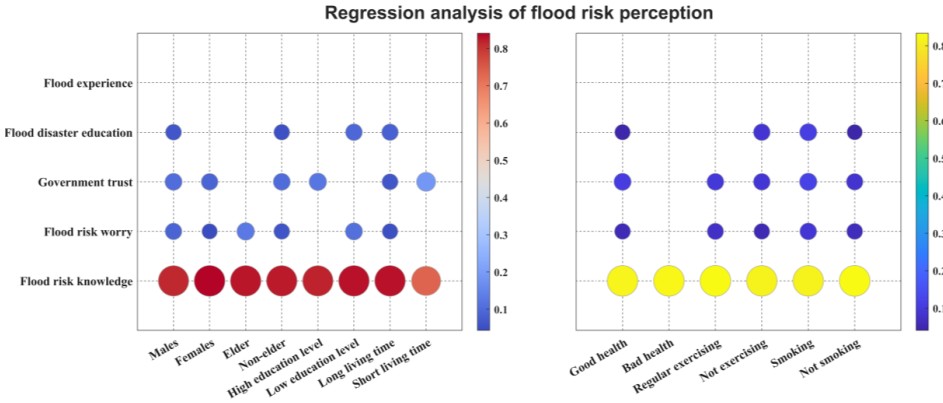

**Fig. 4. Regression analysis of flood risk perception.**
For individuals in good health, only flood risk knowledge significantly affected
risk perception (0.821). Among people in bad health, flood risk knowledge showed a
greater effect (0.824), and flood risk worry, government trust and flood disaster
education also affected risk perception with standardized coefficients of 0.059, 0.107
and 0.046. For individuals who regularly exercised, flood risk knowledge, flood risk
worry, government trust and flood disaster education showed significant positive effects
(0.817, 0.056, 0.091 and 0.090) on risk perception. However, among groups without
exercising, flood risk knowledge, flood risk worry and government trust showed a
lower impact, with standardized coefficients of 0.833, 0.076 and 0.097. For people with
bad habit, flood risk knowledge, flood risk worry, government trust and flood disaster
education showed significant effects (0.815, 0.093, 0.118 and 0.111) on risk perception.
But among groups without bad habit, the effect of flood risk knowledge was lower
(0.831), while flood risk worry, government trust and flood disaster showed a greater



impact on risk perception with standardized coefficients of 0.063, 0.086 and 0.041.

**3.5. Influencing factors of flood preparedness**

**Table 13** listed the stepwise regression results of flood preparedness. Threat
appraisal had a significant and positive influence (0.213), followed by government trust
(0.178), flood risk knowledge (0.140), flood disaster education (0.08) and flood risk
worry (0.07), while only flood experience exhibited a negative effect (-0.09). Lower
influence of threat appraisal on flood preparedness suggested that high risk perception
may be associated with insufficient flood preparedness behavior. This study also
considered socio-economic features as group categories, and explored the influence
effects of different factors (**Fig. 5**). **Supplementary materials** provided more detailed
information about stepwise regression.
In the high risk-perception groups, threat appraisal significantly and positively
affected flood preparedness (0.226), followed by flood disaster education (0.213), flood
risk worry (0.162), government trust (0.123), flood risk knowledge (0.103). Only flood
experience had the negative effect (-0.171). For the low risk-perception groups, threat
appraisal had a higher influence (0.309) on flood preparedness, but other factors were
not significant. Among individuals with low response intention, only threat appraisal
and government trust had significant positive effects on flood preparedness (0.211 and
0.172). For people with high response intention, the effect of threat appraisal and
government trust increased and reached 0.216 and 0.193 respectively, while flood risk
knowledge, flood disaster education and flood experience also exhibited significant
influences (0.217, 0.106 and -0.112). High response intention improved the influence



effect of threat appraisal and government trust and led to significant effects of other
different factors.
**Table 13**
Stepwise regression analysis results of flood preparedness.

| Variable | Standardized coefficients | p-value |
|---|---|---|
| Threat appraisal | 0.213 | 0.000 |
| Flood risk knowledge | 0.140 | 0.000 |
| Flood risk worry | 0.072 | 0.034 |
| Government trust | 0.178 | 0.000 |
| Flood disaster education | 0.075 | 0.032 |
| Flood experience | -0.078 | 0.024 |
| $R^2$ | 0.184 | |
| Adjusted $R^2$ | 0.177 | |
| F | 27.439 | |

Among males, threat appraisal, flood risk knowledge and government trust had
significant effects on flood preparedness (0.263, 0.192 and 0.240). In females, threat
appraisal, government trust, and flood disaster education significantly affected flood
preparedness (0.154, 0.141, and 0.123). The effect of threat appraisal was crucial in
males compared to females. Among the elderly, only threat appraisal and government
trust had significant and positive effects on flood preparedness (0.237 and 0.319). But
in non-elderly individuals, the influence of threat appraisal and government trust was
lower (0.217 and 0.155). Furthermore, flood risk knowledge, flood risk worry, flood
disaster education and flood experience significantly affected flood preparedness, with
standardized coefficient of 0.136, 0.028, 0.096 and -0.086 respectively.

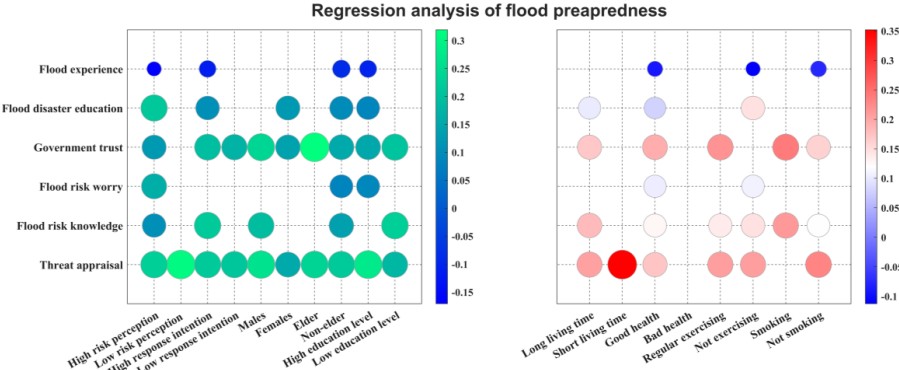

**Fig. 5. Regression analysis of flood preparedness.**

In people with high education background, threat appraisal, flood risk worry, government trust and flood experience significantly affected flood preparedness (0.276, 0.088, 0.152 and -0.102). But among individuals with low education, the effect of threat appraisal and government trust declined and reached 0.180 and 0.205 respectively. Flood risk knowledge also had a positive influence on flood preparedness (0.226). Among individuals with long living time, threat appraisal, flood risk knowledge, government trust and flood disaster education showed significant and positive effects on flood preparedness (0.204, 0.180, 0.169 and 0.102). But for those with short residence duration, only threat appraisal exerted a significant effect (0.352).

For people in bad health, threat appraisal and flood risk knowledge exhibited relatively higher effects (0.602 and 0.292), but none of the variables were statistically significant. Among groups in good health, although only flood experience had the negative effect (-0.091), all variables affected flood preparedness significantly and positively. In people without exercising, threat appraisal, flood risk knowledge and government trust showed significant and positive effects on risk perception (0.207,


0.147 and 0.116). But among groups with regular exercising, the effect of threat
appraisal and government trust improved and achieved 0.208 and 0.218 respectively,
while the influence of flood risk knowledge decreased with standardized coefficients
of 0.137. For individuals without bad habit, threat appraisal, flood risk knowledge and
government trust had significant effects on flood preparedness (0.229, 0.119 and 0.161),
while only flood experience exhibited a negative influence (-0.078). However, among
people with bad habit, the effect of flood risk knowledge and government trust
improved, and both significantly and positively affected flood preparedness (0.210 and

0.238)

**3.6 Influence path of flood preparedness**

This study examined the moderating and mediating effect and explored the

influence path between flood risk perception and flood preparedness. **Supplementary**
**materials** presented more detailed illustration. Risk perception, flood preparedness,
response intention and social-economic factors acted as independent, dependent,
mediating and moderating variables respectively. In **Fig. 6(a)**, health condition played
the negative moderating role between threat appraisal and flood preparedness. Threat
appraisal had significant and positive effects on response intention (0.397) and flood
preparedness (0.313), while response intention also positively influenced flood
preparedness (0.174). Under the influence of health condition and response intention,
the direct effect of threat appraisal on flood preparedness was greater than indirect effect.
The slope of low, medium and high moderation changed obviously and tended to be
gentle in **Fig. 7(a)**. With the increasing moderation effect, health condition interfered
with the influence of threat appraisal on flood preparedness. In **Fig. 7(b),** as health
condition worsened (M-1SD), threat appraisal exhibited a significant and positive
prediction effect on flood preparedness (Slope =0.400). The prediction effect of threat
appraisal gradually weakened with improved health condition. Threat appraisal showed
a positive prediction effect (Slope =0.238), as health condition became good (M+1SD).
Improvement in health condition reduced the positive effect of threat appraisal on flood
preparedness.

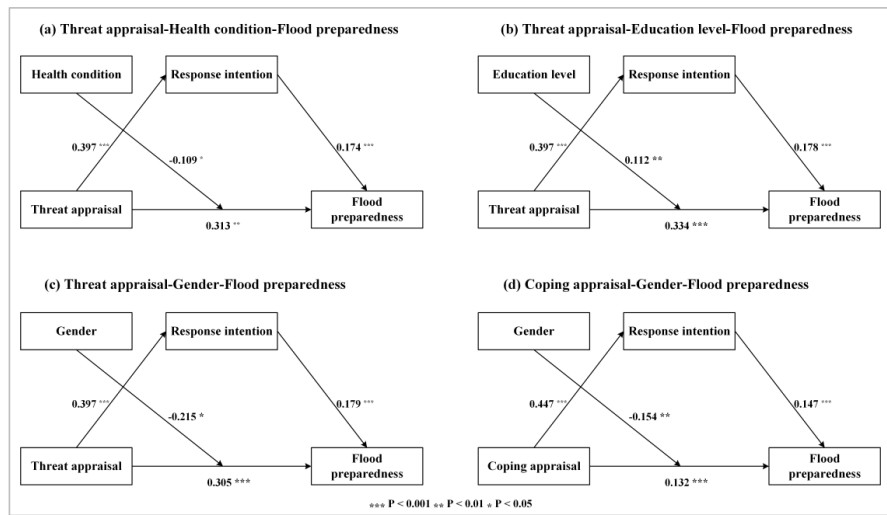


**Fig. 6. Influence path of flood preparedness.**
Relationship between threat appraisal and flood preparedness was positively
moderated by education level. Threat appraisal showed significant and positive effects
on response intention and flood preparedness (0.334) in **Fig. 6(b)**. Response intention
also demonstrated a positive effect on flood preparedness (0.178). Direct effect of threat
appraisal on flood preparedness was greater than indirect effect under the impact of
education level and response intention. Slope test revealed that, in **Fig. 7(b)**, when





education level was low (M-1SD), threat appraisal showed a positive prediction effect
on flood preparedness (0.211). When education level was high (M+1SD), threat
appraisal also significantly and positively predicted flood preparedness with greater
prediction effect (0.457). As education level improved, there was an ascending trend in
the predictive effect of threat appraisal.

Gender also played the negative moderating effect between threat appraisal and

flood preparedness **in Fig. 6(c)**. Threat appraisal exhibited positive effects on response
intention and flood preparedness (0.305), and response intention also had a positive
effect (0.179). With the influence of gender and response intention, direct effect of
threat appraisal on flood preparedness was more substantial than indirect effect. In **Fig.**
**7(c),** for individuals with male gender (M-1SD), threat appraisal positively predicted
flood preparedness (0.426). For individuals with female gender (M+1SD), threat
appraisal positively still showed a significant and positive prediction effect (0.211).
Predictive effect of threat appraisal on flood preparedness was essential in the male
group compared to females.

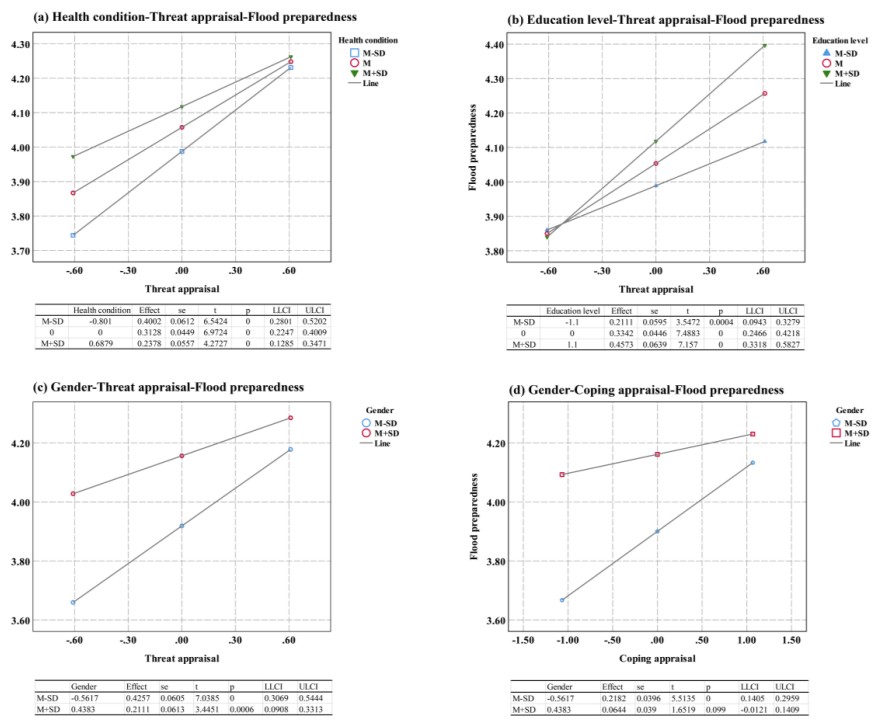


**Fig. 7. Moderating effect on flood preparedness.**

Gender negatively moderated the relationship between coping appraisal and flood
preparedness. **In Fig. 6(d)**, coping appraisal positively influenced response intention
(0.447) and flood preparedness (0.132). Response intention also showed a positive
effect on flood preparedness (0147). Under the influence of gender and response
intention, coping appraisal exhibited a greater direct effect on flood preparedness than
indirect effect. In **Fig. 7(d),** when gender was male (M-1SD), coping appraisal
positively predicted flood preparedness (0.218). When gender was female (M+1SD),
coping appraisal represented a positive but insignificant prediction effect (0.064).
Coping appraisal had a lower predictive effect on flood preparedness among females.

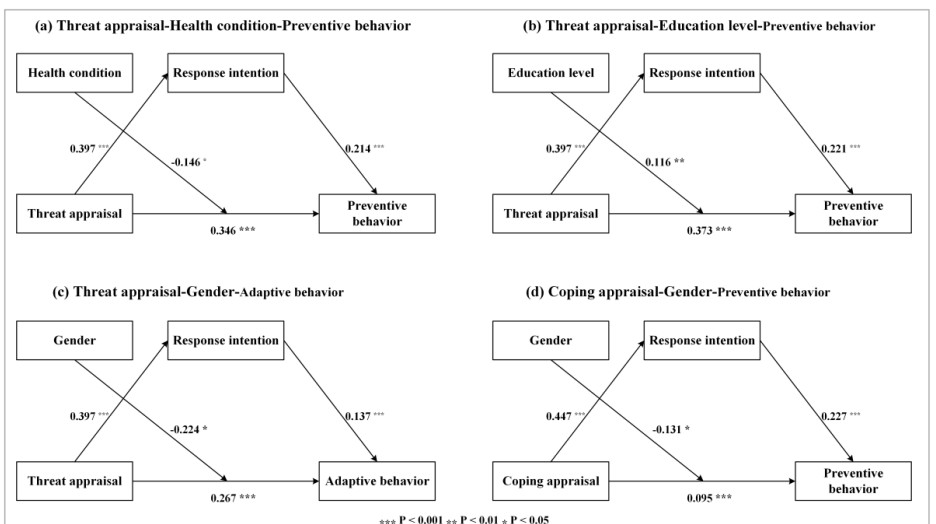

**Fig. 8. Influence differences on adaptive and preventive behavior.**

This study also explored the behavior differences of flood preparedness influenced by flood risk perception, response intention and social-economic factors. Health condition played a negative mediating effect between threat appraisal and preventive behavior, and response intention showed the moderation effect in **Fig. 8(a)**. Threat appraisal could transform into preventive behavior under the influence of response intention and health condition. Slope test (**Fig. 9(a)**) revealed that prediction effect between threat appraisal and preventive behavior diminished with improved health condition. Additionally, education level displayed a moderating effect between threat appraisal and preventive behavior (**Fig. 8(b)**). Threat appraisal could transform into preventive behavior under the fluence of education level and response intention. But prediction effect reduced as education level increased based on slope test (**Fig. 9(b)**).

Gender played a moderation effect between threat appraisal and adaptive behavior. Threat appraisal could transform into adaptive behavior with the effect of response and



gender **(Fig. 8(c))**. When gender was male (M-1SD), threat appraisal demonstrated a
stronger positive prediction effect on adaptive behavior (0.458) in **Fig. 9(c)**.
Furthermore, coping appraisal could transform into preventive behavior under the
mediating effect of response intention and the moderation effect of gender **(Fig. 8(d))**.
When gender was male (M-1SD), coping appraisal predicted positively preventive
behavior (0.168) in **Fig. 9(d)**. When gender was female (M+1SD), coping appraisal had
a weak and statistically insignificant prediction effect on preventive behavior (0.0378).
Risk perception was more likely to be translated into preventive behavior among males.

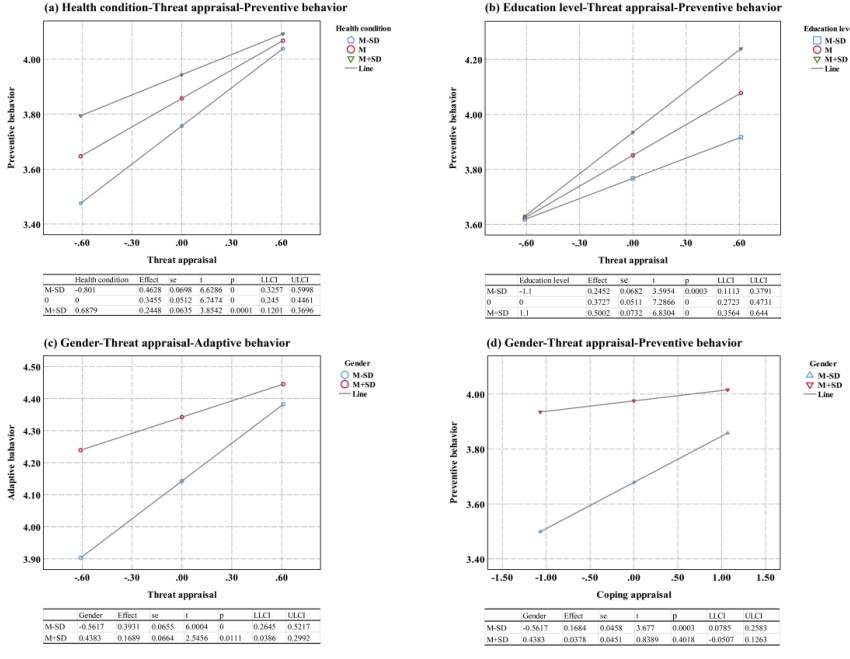

**Fig. 9. Moderating effect on adaptive and preventive behavior.**
**4. Discussion**
This study revealed no significant difference in risk perception between genders,
but females demonstrated a higher level of flood preparedness, consistent with previous





research (Rana et al., 2020; Rasool et al., 2022). Individuals who regularly exercised
exhibited higher risk perception, because adequate physical activity enhanced their
response and judgment capabilities, and thereby made cognitive activities more active.
The elderly, particularly those aged 51-60 and above 60, showed higher risk perception
yet lower flood preparedness. Often regarded as socially vulnerable groups, the elderly
were more probably perceived higher risk (Harlan et al., 2019), and due to insufficient
energy and reaction, they struggled to undertake practical behaviors in response to
hazards. Higher risk perception was observed on groups with low education level, while
those with high education level showed a higher level of flood preparedness. People
with lower educational degree, typically associated with lower social status, were more
inclined to engage in occupations that are dangerous or risky (Bollettino et al., 2020;
Kiani, Najam, & Rana, 2022), which incentivized them to proactively perceive flood
risks. But highly educated people sought diverse information about disasters and
prepared adequately for floods (Rana et al., 2020). Long living time made people
become acquainted with local conditions, leading to a positive perception of flood risk.
People who experienced and worried about flood displayed the higher risk perception
and made adequate preparation for floods in our findings. Past flood experiences tended
to trigger risk perception and a greater intention to take adjustment action (Ao et al.,
2020). Individuals were more likely to declare higher risk perception and preparedness,
when floods were associated with negative emotions or memories (Rufat & Botzen,

2022).

Enough high threat appraisal could trigger coping appraisal (Schlef et al., 2018),



which both caused the higher protection motivation and promoted the mitigation
measures (Kurata et al., 2022). Despite finding the high threat appraisal and the medium
coping appraisal, threat appraisal might not reach the necessary threshold that
effectively triggered coping appraisal, and coping appraisal showed no effect on flood
preparedness in our results. Individuals tended to rely on threat appraisal to perceive
risk and failed to generate an adequate coping appraisal, leading to insufficient risk
perception. Therefore, risk perception struggled in translating into flood preparedness,
resulting in the unbalanced relationship with flood preparedness. The influence of threat
appraisal on flood preparedness was greater in groups with low risk perception
compared to those with high risk perception. The transformation of low risk perception
into flood preparedness could be attributed to the relatively stronger effect of threat
appraisal on flood preparedness. The association between high risk perception and low
flood preparedness could arise from the weak effect of threat appraisal on flood
preparedness. However, due to the significant influence of other factors, such as
government trust, individuals were more likely to be better prepared for floods among
groups with high risk perception.

Various social-economic characteristic influenced individuals' preferences for

different ways to achieve risk perception and flood preparedness, based on regression
analysis. Females had higher flood worry and depended more on flood knowledge to
perceive risk than males, possibly owing to the general cognition that women were
more vulnerable and sensitive (Eryılmaz Türkkan & Hırca, 2021). But flood worry
showed lower effect on risk perception than that in males. It was suggested that females





should keep calm, and improve risk perception through flood knowledge. The elderly
relied on both flood knowledge and worry for risk perception. Although they exhibited
a greater influence of government trust on flood preparedness, lower level of
government trust could potentially hinder their flood preparedness efforts. People with
low education level preferred flood knowledge for risk perception, and were advised to
bolster their trust in government to improve flood preparedness. Individuals with longer
residency durations relied more on flood knowledge for risk perception, while those
with short living time, unfamiliar with local floods, depended more on government trust
for risk perception and favored threat appraisal to achieve flood preparedness. Groups
with poor health relied more on flood knowledge for flood preparedness, as adequate
risk knowledge could compensate for physical functional limitations. Individuals with
regular exercising group showed a preference for threat appraisal in preparation for
floods. Moreover, individuals with bad habits, considered psychologically fragile and
sensitive, preferred flood risk worry and knowledge, and government trust for risk
perception.

In our study, risk perception, including both threat and coping appraisal,

demonstrated a direct influence on flood preparedness, and response intention also
exhibited a mediating effect. Socio-economic factors, especially education level and
health condition, played a moderating effect between risk perception and flood
preparedness. People with high education level could better deal with complicated
information and act promptly during the time lag between action and outcome (Dootson
et al., 2022). As health condition improved, there was a negative predictive effect of





threat appraisal on flood preparedness. Though people reporting good health displayed
confidence with physical function, overconfidence could hinder the translation of risk
perception into preparedness (Bollettino et al., 2020), and these groups should attach
importance to timely feedback in response to floods. Among males, despite the low
level of flood preparedness, threat and coping appraisal were stronger predictors of
flood preparedness. With the effect of response intention and socio-economic factors,
risk perception could transform into flood preparedness, and caused the differences of
preventive and adaptive behaviors. People with high education level would more
probably perceive risk and engage in preventive behavior against flooding. Groups with
bad health were more likely to perceive flood risk, and adopt preventive measures.
This study revealed the influence of socio-economic factors on risk perception and
flood preparedness. But we only found the influence path in part of factors and results
may not be generalized in all the socio-economic characteristics. Rationality and
reliability of influence path need further empirical validation in future studies. With the
climate change, the adoption of different behaviors was significantly influenced by how
individuals perceived and evaluated risk (Bodoque, Díez-Herrero, Amerigo, García, &
Olcina, 2019). When risk events were associated with adequate benefits, individuals
tended to exhibit a preference for adaptive behaviors (Keshun Zhang et al., 2021).
Consequently, comprehensive analysis of benefits and costs was crucial in
understanding risk perception and preparedness.
**5. Conclusion**
We designed and conducted a questionnaire survey to explore influence



relationship between risk perception and flood preparedness. Participants exhibited the
medium perception of food risk and demonstrated higher flood preparedness. High
levels of risk perception were observed on groups of regular exercising, the elderly,
flood experience, low education level, long living time and flood worry. Higher floods
preparedness was more prevalent among groups of females, the elderly and high
education level. Individuals tended to rely predominantly on threat appraisal to perceive
risk, and failed to trigger the adequate coping appraisal. This process resulted in a
challenging translation of risk perception into flood preparedness, accompanied with
unbalanced relationship. Groups with distinct social-economic features showed
different preferences to realize risk perception and flood preparedness. Females relied
more on flood knowledge to perceive risk and were suggested to keep calm and enhance
risk perception through flood knowledge. Elderly individuals and people with low
education level also depended on flood knowledge for risk perception, while lower
government trust possibly hindered taking flood preparedness. Path analysis indicated
that threat appraisal could transform into flood preparedness, influenced by response
intention, education level, or health status condition. Groups with high education level
or bad health would more probably perceive risk and engage in preventive behavior.
This study provided essential strategies for promoting flood preparedness in response
to floods. Future research should consider the benefits and costs associated with risk to
reveal the heterogeneity of preparedness behaviors.
**Author contribution**
**Yabo Li**: Methodology, Investigation, Writing - Original Draft. **Peng Wang**:



Conceptualization, Writing - Review & Editing, Supervision.
**Declaration of interests**

The authors declare that they have no known competing financial interests or

personal relationships that could have appeared to influence the work reported in this
paper.
**Acknowledgements**

This work was supported by the National Natural Science Foundation of China

[Grant No. 51908249], the Natural Science Foundation of Jiangsu Province [Grant No.
SBK2023022191], the Natural Science Foundation of the Jiangsu Higher Education
Institutions of China [Grant No. 19KIB560012], the High-level Scientific Research
Foundation for the introduction of talent for Jiangsu University [Grant No. 18JDG038],
the Innovative Approaches Special Project of the Ministry of Science and Technology
of China [Grant No. 2020IM020300], and the Science and Technology Planning Project
of Suzhou [Grant No. ST202218].

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
