# Peer review of "Regression analysis of flood risk perception"

_Natural Hazards and Earth System Sciences, 2024_

## Author Comment (AC1)

**Details of reply**

Dear reviewers,

We would like to thank the reviewers for his/her interest in our work for their effort, constructive criticism and suggestion. We appreciate the insightful comments, as these would contribute to improving the manuscript's robustness and quality. We provide a point-by-point reply to the general and specific comments raised as follows:

**REVIEWER 1:**

In the manuscript, Li and Wang have developed a survey to measure flood risk perceptions and flood preparedness. They have used statistical tests and regression analysis on the survey responses to determine the effect of different variables, e.g., gender and education level, on these dependent variables. These kind of surveys and studies are highly relevant and useful for informing disaster risk education, and risk mitigation policies.

**Reply: Thanks greatly, we appreciate your positive assessment of our study.**

I have some major concerns regarding the comprehensiveness of authors' methodologies that make it difficult to validate some of their conclusions. I have listed both my major and other concerns below.

1.The language in the paper could be improved for readability. Especially in the abstract and introduction, the use of past tense makes it harder to follow the definitions and literature review. To be clear, the improvement in language is not a reflection on the quality of the scientific content of this paper, which was reviewed independently.

**Reply: Thank you for your suggestions, and we would enhance the overall readability of this paper. We would consider the use of present tense in the section of abstract and introduction, rephrasing them for better readability.**

2.The methods used in the paper are only mentioned but not described. While it is okay to use standard methods like linear regression without explaining them, it will be helpful to the audience of this journal to explain when special methods are used such as stepwise regression. Additionally, references to these methods are consistently missing. Additional comments below mention some of the methods for which it would be helpful to add descriptions and references.

**Reply: Thank you for these helpful suggestions. Stepwise regression is a type of multiple linear regression that can select the best-fitted combination of independent variables for dependent variable prediction with forward-adding and backward-deleting variables. The stepping procedure begins as an initial model definition, with a stepped forward addition of a variable to the previous model. The critical F value is then used to check the eligibility of the added variable. With a new variable added, the previous variables in the model may lose their predictive ability. Thus, stepping criteria are used to check the significance of all the included variables. If the variable is insignificant, then the backward method is used to delete it. Forward selection and backward elimination are repeated until no variable is added or removed. Stepping procedure would stop until the optimized model is established. We mainly considered the way of back elimination in stepwise regression and would add more details about stepwise regression analysis in the paper. And we would add more detailed description and relevant references in this paper.**

3.Since the primary contribution of this manuscript is reviewing survey responses, it will be helpful

to understand how the 844 respondents were selected across the geographic region, and the criteria for selection. Please provide additional information, such as, why 844 respondents were selected instead of more or fewer.

**Reply: Thanks a lot, we used the method of random sampling to perform the survey. Random sampling is a type of probability sampling method where everyone is selected entirely on a chance, with each one having an equal probability of being chosen. The reason for adopting random sampling was to reduce selection biases in the survey and to be able to calculate sampling error. There are various methods mentioned in the literature to determine the sample size. The sample size of respondents was calculated using Yamane's formula. Using a confidence level at 95%, the sampling method proposed sample size of 844. Random sampling was chosen to carry out the survey. A total of 737 samples were finalized after discarding the incomplete questionnaires.**

$$n = \frac{N}{1 + N(e)^2}$$

**where n is the sample size, N denotes the size of the resident population, and e is the precision level.**

4.In addition to the above, please provide additional statistics to ascertain whether the survey respondents were reflective of the socio-economic distribution of the targeted geographic region.

**Reply: Thanks a lot for your useful suggestions. In our survey, women accounted for 56.2% and men accounted for 43.8%. The number of people with a bachelor's degree or above accounted for 58.1% of the total number of the survey respondents, and the rest were high school education or below, accounting for 41.9%. The age groups were: 26 to 40 years old (33.3%), 41 to 60 years old (25.9%), and 9 people over 60 years old (11.9%). The flood control knowledge survey[1] issued by Nanjing Municipal People's Government included the social and economic distribution of citizens in recent years, which was consistent with these socio-economic features of survey respondents in our study. And we would add the related illustration in this paper.**

5.The authors have mentioned that a detailed description of the survey questionnaire is available in supplementary material. Given the high relevance of the survey development to the authors' conclusions, it will be helpful if some more detailed information regarding survey development, improvement, and questions was included in the main text.

**Reply: Thank you for your suggestions. To identify the potential problems (e.g. unclear and ambiguous questions), preliminary tests of online questionnaire were conducted before officially issued for the survey. We collected and sorted out the survey results of interviewees and modified the questionnaire reasonably according to the feedback. Then we deleted and reduced the questionnaire options that may lead to bias and misunderstanding. These details would be added to this paper.**

6.For the purpose of drawing conclusions from the surveys, all 737 valid responses are considered ground truth and representative of the entire population of the targeted geographic region. In order to substantiate the conclusions, it will be beneficial if sensitivity analysis was performed on results. One possible approach could be providing confidence intervals on the results and regression coefficients. Another approach could be bootstrapping where a subsample of responses are selected
* * *
[1] More information was shown in this website.
https://www.nanjing.gov.cn/hdjl/zjdc/wsdc/dcbg/202310/t20231012_4030100.html

multiple times, and the coefficients are generated. This would help quantify the variability of the results in order to determine whether the differences across various factors are within the error margins or not, in order to draw conclusions.

**Reply: Thanks greatly for your useful suggestions. And we would add the confidence interval on the results and regression coefficients in this paper.**

7.Most of the text in Section 3 lists the numbers already present in the respective tables. The section could be made more succinct by only including the key observations from the tables.

**Reply: Thanks greatly for your valuable suggestions. And we would modify this section carefully and make it more concise and accessible.**

8.Line 163 - Were responses also collected from online distribution of the survey and did those responses match the ones collected directly by interviewers?

**Reply: Thanks a lot, we tested the online survey to identify the potential problems, such as the unclear and ambiguous questions in the questionnaire. Based on the feedback from respondents, we could improve the questionnaire reasonably and reduce the questionnaire options that may lead to bias. Because there was the different structure and design between the original online questionnaire and face-to-face questionnaire surveys, it is difficult to determine whether these responses are well matched. And all the analysis in this study was based on the results of face-to-face questionnaire surveys.**

9.Line 168 - It will be helpful to include whether any analysis was done to review interviewer bias in the responses.

**Reply: Thanks for your useful suggestions. When designing and improving the questionnaire, we considered the order and logical relationship of the questionnaire questions to avoid subjective bias and misleading questions. And interviewer received excellent survey skills training, such as anonymous survey, before formal interview to avoid the misunderstanding and confusion during the interview. Therefore, we didn't carry out the analysis about interviewer bias.**

10.Line 179 - Please include information about how the valid and invalid responses were determined.

**Reply: Thanks a lot, and we mainly excluded the invalid responses following the criteria. (1) Incomplete questionnaire, that is, a considerable part of the questionnaire was not filled in. (2) Respondents did not understand the contents of the questionnaire and answered the questionnaire incorrectly, or did not answer the questionnaire according to the requirements of the guidance. (3) Interviewees answered the questionnaire with no change, such as in the five-point Likert Scale, no matter what the question, respondents will choose the same answer all through. (4) Defective questionnaires, that is, several pages are missing or cannot be identified. (5) Inconsistent or obviously wrong questionnaires.**

11.Lines 184-187 - It will be helpful to provide more information about the implementation of Mann-Whitney U and Kruskal-Wallis statistical tests, along with relevant references. The current brief description is not sufficient to understand why these tests were chosen, how they were implemented, and their objectives.

**Reply: Thank you for these helpful suggestions. The Mann-Whitney U statistical test is a nonparametric statistical test used to compare the values of a variable between two independent groups. The Kruskal-Wallis statistical test is also a nonparametric statistical test used to compare the values of a variable between several independent groups. The Mann-Whitney U test was used for 'yes or no' questions, and the Kruskal-Wallis test was for**

questions with three or more answer choices. These tests were used to compare the differences of flood risk perception and flood preparedness between two and several independent groups in this study. If the Mann-Whitney U statistical test returns a P-value less than 0.05, the compared categories are significantly different. As a result, they can be ranked using the Mann-Whitney U statistical test's mean rank. If the P-value of the Kruskal-Wallis statistical test is less than0.05, the difference between the compared categories is considered significant. These statistical tests are frequently used and performed via statistical software. Accordingly, these statistical tests were conducted in this study using SPSS. And we would add more detailed description and relevant references in this paper.

12.Line 191 - Briefly describe stepwise regression and provide relevant references.

Reply: Thank you for your suggestions. Stepwise regression is a type of multiple linear regression that can select the best-fitted combination of independent variables for dependent variable prediction with forward-adding and backward-deleting variables. And we would add more detailed description and relevant references in this paper.

13.Line 192 - What is Model 5?

Reply: Thanks a lot for your advice. Model 5 is a moderated mediation model. In this model, risk perception, flood preparedness, response intention and social-economic factors acted as independent, dependent, mediating and moderating variables respectively. And we would modify this illustration in the paper.

14.Line 198 - What are Cronbach's α and KMO values? Please provide brief descriptions along with relevant references.

Reply: Thank you for your suggestions. Cronbach's Alpha coefficient is the most widely used measurement method in the test of questionnaire reliability. This coefficient is a number distributed between 0 and 1. If it is greater than 0.7, the data is acceptable. If it is greater than 0.8, the data is valuable. Cronbach's Alpha coefficient greater than 0.9 indicates high internal consistency. The calculation of KMO value is a common method in the test of questionnaire validity. The KMO value is used to evaluate the correlation between the variables in the data sample. KMO value is very important for the test of questionnaire validity. If the KMO value is greater than 0.7, it that the questionnaire has good validity and can be used for factor analysis and statistical analysis. If the KMO value is less than 0.5, it may mean that there is a problem in the design of questions in the questionnaire, and the questionnaire needs to be corrected or reconstructed. And we would add more detailed description and relevant references in this paper.

15.Line 232 - What is mean rank?

Reply: Thanks a lot for your suggestions. If the Mann-Whitney U statistical test returns a P-value less than 0.05, the compared categories are significantly different. As a result, they can be ranked using the Mann-Whitney U statistical test's mean rank. If the P-value of the Kruskal-Wallis statistical test is less than0.05, the difference between the compared categories is considered significant.

16.Section 3.2 - Why was Mann-Whitney U test used for binary variables and the Kruskal-Wallis test used for multi-category variables? Can the rank values between the two tests be compared with each other?

Reply: Thanks a lot for your suggestions. The Mann-Whitney U statistical test is a nonparametric statistical test used to compare the values of a variable between two

independent groups. The Kruskal-Wallis statistical test is also a nonparametric statistical test used to compare the values of a variable between several independent groups. The Mann-Whitney U test was used for 'yes or no' questions, and the Kruskal-Wallis test was for questions with three or more answer choices. These tests were used to compare the differences of flood risk perception and flood preparedness between two and several independent groups in this study. Because the test categories of each variable were different in these two nonparametric statistical tests, the rank values between the two tests could not be compared with each other.

17. Fig 3 - The colors in the correlation plot do not appear to match the color bar. For example, the diagonal should be solid red since correlation=1.0, but it's white indicating correlation=0. Similarly, value of 0.05 is shaded light red while a value of 0.76 is shaded white. As a result, Section 3.3 could not be reviewed for accuracy, and its conclusions could not be substantiated.

**Reply: Thanks greatly for your valuable suggestions. We would check and redraw this figure and modify the overall organization of Section 3.3 carefully.**

18. Section 3.4 - Briefly describe models 1, 2, and 3. From Table 12, it appears that each model used a different set of features.

**Reply: Thanks for your suggestions. We used stepwise regression to reveal the influencing factors of flood risk perception. First, we selected all variables for regression analysis in model 1, and found that flood risk knowledge showed the significant and positive effect while other variables exhibited relatively lower effects. And then after removing socio-economic variables, we built the model 2 with a high goodness of fit (adjusted $R^2$=0.788). Flood risk knowledge also maintained a higher influence (0.827) on flood risk perception. Furthermore, we excluded the variable flood risk knowledge in model 3, with a low goodness of fit (adjusted $R^2$=0.246). But government trust, flood experience, flood disaster education and flood risk worry significantly and positively influenced risk perception, indicated by increased regression coefficients. The effect of flood experience on flood risk perception shifted from insignificant to significant. We found that although flood risk knowledge could significantly promote risk perception, it also inhibited and decreased the positive effects of other factors. And we would add this brief and concise description.**

19. Sections 3.4, 3.5 - The difference in regression coefficients between various groups (e.g., males and females) appear quite small. It will be helpful to see whether these differences are in fact indicative of reality or within the error bars (such as, confidence intervals) based on the number of survey responses.

**Reply: Thanks a lot for your suggestions. Although some regression coefficients are small in different groups, this study aimed to explore the differences in the influencing factors of flood risk perception and flood preparedness between different groups. It is more important that whether these different influencing factors show the significant effect or not. These small difference in regression coefficients between various groups could be accepted in this study. And we would also add the confidence intervals of different variables in this paper.**

20. Fig 4 - Why are certain coefficients missing from the figure, e.g., females and flood disaster education? Same for Fig 5.

**Reply: Thanks a lot for your suggestions. We only listed the significant results of regression analysis in Fig. 4 and Fig. 5. And therefore, some insignificant coefficients were not presented in these figures, which we mentioned in this original manuscript (Line 330). We would**

**highlight this description about Fig. 4 and Fig. 5 in the paper again.**

21.Section 3.6 - Please describe how the influence path analysis was implemented, along with relevant references.

**Reply: Thanks a lot for your suggestions. We performed a moderated mediation model in PROCESS macro program of SPSS to capture the influence path between flood risk perception and flood preparedness. The PROCESS program can effectively test the moderated mediation model and help to clarify the mediating and moderating roles of different variables. All statistical analyses were conducted at a significance level of 0.05. In this model, risk perception, flood preparedness, response intention and social-economic factors acted as independent, dependent, mediating and moderating variables respectively. And we would add more detailed description and relevant references in this paper.**

22.Section 3.6 - New taxonomy is presented, e.g, M-1SD, without any explanation for its meaning. As a result, it was not clear how to interpret the figures and results.

**Reply: Thanks a lot for your suggestions. M-1SD means that the value of a variable is one standard deviation below the mean value. We aimed to explore the moderating effect among independent, dependent, moderating variables by increasing and decreasing the level of moderating variable. In this way, we could reveal that whether the independent variable has a significant positive predictive effect on the dependent variable or not, with moderating variable being one standard deviation below (M-1SD) or above (M+1SD) its mean value. And we would modify this section carefully and make it more concise and accessible.**

---

## Author Comment (AC2)

**Details of reply**

Dear reviewers,

We would like to thank the reviewers for his/her interest in our work for their effort, constructive criticism and suggestion. We appreciate the insightful comments, as these would contribute to improving the clarity our manuscript. We provide a point-by-point reply to the general and specific comments raised as follows:

**REVIEWER 2:**

Thank you for the opportunity to review the manuscript. The paper needs some improvement before it can be considered for publication. Here are my comments:

1. avoid the word "natural disasters". Disasters are not natural. please refer to IPCC or UNDRR reports.

**Reply: Thank you for your suggestions, and we would avoid this expression in the paper.**

2. The LR needs improvement. Please see Rana et al. 2020. Characterizing flood risk perception in urban communities of Pakistan. The paper reviews theories on flood risk perception. See some more references at the end for improving this section.

**Reply: Thanks a lot, we would consider these references carefully and modify this section better in this paper.**

3. Why is district taken as a socioeconomic factor? Maybe drop it from the analysis? Mean of district doesn't make sense too.

**Reply: Thanks a lot, and this study focused on the urban center of Nanjing including six districts: Gulou, Xuanwu, Jianye, Qinhuai, Qixia and Yuhuatai district respectively. Based on Nanjing Statistical Yearbook, there were socioeconomic differences in these districts, and therefore, we considered it as a socioeconomic factor and mainly used it to reveal the socioeconomic differences of survey respondents in the regional distribution.**

4. Figure 3. Some values of significant correlation are missing in the figure.

**Reply: Thanks greatly for your valuable suggestions. We would check again and redraw this figure 3 carefully.**

5. Please add a model of fitness for regression results.

**Reply: Thanks a lot for your suggestions. The R-square value is a statistic that measures the goodness of fit of a regression model and indicates how well the regression model fits the observed values. The R square value ranges from 0 to 1, and the greater the R square value, the better the regression model fits the observed value. The adjusted $R^2$ is the correction of $R^2$, and the adjusted R2 considers the number of independent variables and the influence of sample size to avoid the problem of over-fitting. RMSE is the most commonly used evaluation model index in regression models. The closer the RMSE value is to 0, the better the model fitness. We would add more descriptions about the fitness of regression analysis in this paper.**

6. The manuscript is too long, maybe cut down on Mann, Kruskal-Wallis tests etc. Regression is the main thing in this paper.

**Reply: Thanks a lot for your advice and correction. Mann-Whitney U and Kruskal-Wallis tests were used to compare the differences of flood risk perception and flood preparedness between variable groups in this study. And we would make this section more concise.**

Overall, the paper is technically sound but needs a little improvement in language and flow. Minor

revisions are suggested.

Minor comments:

1. Need grammar check. Especially figures and abstracts.

**Reply: Thanks a lot for your advice and correction. We will check the grammar, improve the language and flow again and adjust the figures and abstracts in this paper.**

2. L16 Flood, not food

**Reply: Thanks for this correction, and we will adjust this word accordingly.**

3. Figure 5. Flood preparedness. Check spelling.

**Reply: Thanks for your correction, and we will adjust this figure accordingly.**

References to consult:

- https://doi.org/10.1016/j.ijdrr.2016.08.028
- https://doi.org/10.1016/j.ijdrr.2019.101427
- https://doi.org/10.1016/j.jenvman.2022.115309
- https://doi.org/10.1080/17477891.2023.2220947

I wish the authors well with the revision. Good luck.

**Reply: Thanks a lot, we appreciate your positive assessment of our study and will consider these references in this paper.**

---

## Author Response (AR1)

**Details of modification**

Dear reviewers,

We would like to thank the reviewers for his/her interest in our work for their effort, constructive criticism and suggestion. We appreciate the insightful comments, as these would contribute to improving the manuscript's robustness and quality. We provide a point-by-point reply to the general and specific comments raised as follows:

**REVIEWER 1:**

In the manuscript, Li and Wang have developed a survey to measure flood risk perceptions and flood preparedness. They have used statistical tests and regression analysis on the survey responses to determine the effect of different variables, e.g., gender and education level, on these dependent variables. These kind of surveys and studies are highly relevant and useful for informing disaster risk education, and risk mitigation policies.

**Reply: Thanks greatly, we appreciate your positive assessment of our study.**

I have some major concerns regarding the comprehensiveness of authors' methodologies that make it difficult to validate some of their conclusions. I have listed both my major and other concerns below.

1. The language in the paper could be improved for readability. Especially in the abstract and introduction, the use of past tense makes it harder to follow the definitions and literature review. To be clear, the improvement in language is not a reflection on the quality of the scientific content of this paper, which was reviewed independently.

Reply: Thank you for your suggestions, and we would enhance the overall readability of this paper. We have considered the use of present tense in the section of abstract and introduction, rephrasing them for better readability. This modification is shown in the revised paper (page1-5, line 10-115).

2. The methods used in the paper are only mentioned but not described. While it is okay to use standard methods like linear regression without explaining them, it will be helpful to the audience of this journal to explain when special methods are used such as stepwise regression. Additionally, references to these methods are consistently missing. Additional comments below mention some of the methods for which it would be helpful to add descriptions and references.

Reply: Thank you for these helpful suggestions. Stepwise regression is a type of multiple linear regression that can select the best-fitted combination of independent variables for dependent variable prediction with forward-adding and backward-deleting variables. The stepping procedure begins as an initial model definition, with a stepped forward addition of a variable to the previous model. The critical F value is then used to check the eligibility of the added variable. With a new variable added, the previous variables in the model may lose their predictive ability. Thus, stepping criteria are used to check the significance of all the included variables. If the variable is insignificant, then the backward method is used to delete it. Forward selection and backward elimination are repeated until no variable is added or removed. Stepping procedure would stop until the optimized model is established. We mainly considered the way of back elimination in stepwise regression and would add more details about stepwise regression analysis in the paper. And we added more detailed description and relevant references in the revised paper (page 22, line 202-210).

Stepwise regression is a type of multiple linear regression that identifies the optimal combination of independent variables for predicting the dependent variable (Chen et al., 2013; Wang et al., 2023), including forward-adding and backward-deleting methods. When a new variable is introduced, previous variables in the model may lose predictive ability. Then the backward method is employed to remove the new and insignificant variable. The stepwise procedure is terminated once the optimized model is established. We mainly considered the backward elimination approach in stepwise regression and revealed the impact of different factors on risk perception and flood preparedness.

3. Since the primary contribution of this manuscript is reviewing survey responses, it will be helpful to understand how the 844 respondents were selected across the geographic region, and the criteria for selection. Please provide additional information, such as, why 844 respondents were selected instead of more or fewer.

Reply: Thanks a lot, we used the method of random sampling to perform the survey. Random sampling is a type of probability sampling method where everyone is selected entirely on a chance, with each one having an equal probability of being chosen. The reason for adopting random sampling was to reduce selection biases in the survey and to be able to calculate sampling error. There are various methods mentioned in the literature to determine the sample size. The sample size of respondents was calculated using Yamane's formula. Using a confidence level at 95%, the sampling method proposed sample size of 844. Random sampling was chosen to carry out the survey. A total of 737 samples were finalized after discarding the incomplete questionnaires. This modification is shown in the revised paper (page 9-10, line 176-188).

The sample size of respondents was calculated using Yamane's formula (Rasool et al., 2022). A sample size of 844 was proposed using the sampling method with 95% confidence level, and random sampling was chosen to conduct the survey. This study mainly excluded the invalid responses following the criteria: (1) Incomplete questionnaire, that is, a considerable part of the questionnaire was not filled in. (2) Respondents did not understand the questionnaire and answered incorrectly or did not answer according to the guidance. (3) Interviewees chose the same answer all through even if the question changed. (4) Some questionnaires were missing pages or could not be identified. (5) Inconsistent or obviously wrong questionnaires. Eventually, this study distributed 844 questionnaires and obtained 737 valid questionnaires after excluding 107 invalid ones with an effective rate of 87.32%.

$$n = \frac{N}{1 + N(e)^2} \tag{1}$$

where n is the sample size, N is the resident population, and e is the precision level.

4.In addition to the above, please provide additional statistics to ascertain whether the survey respondents were reflective of the socio-economic distribution of the targeted geographic region. Reply: Thanks a lot for your useful suggestions. In our survey, women accounted for 56.2% and men accounted for 43.8%. The number of people with a bachelor's degree or above accounted for 58.1% of the total number of the survey respondents, and the rest were high school education or below, accounting for 41.9%. The age groups were: 26 to 40 years old (33.3%), 41 to 60 years old (25.9%), and 9 people over 60 years old (11.9%). The flood control

knowledge survey1 issued by Nanjing Municipal People's Government included the social and economic distribution of citizens in recent years, which was consistent with these socio-economic features of survey respondents in our study. We added the related illustration in Supplementary material.

5. The authors have mentioned that a detailed description of the survey questionnaire is available in supplementary material. Given the high relevance of the survey development to the authors' conclusions, it will be helpful if some more detailed information regarding survey development, improvement, and questions was included in the main text.

Reply: Thank you for your suggestions. To identify the potential problems (e.g. unclear and ambiguous questions), preliminary tests of online questionnaire were conducted before officially issued for the survey. We collected and sorted out the survey results of interviewees and modified the questionnaire reasonably according to the feedback. Then we deleted and reduced the questionnaire options that may lead to bias and misunderstanding. These details were added to this revised paper (page 9, line 157-164).

To solve the potential problems including unclear and ambiguous questions, preliminary tests of online questionnaire were conducted before officially issued for the survey. We collected and sorted out the survey results of interviewees, and modified the questionnaire reasonably according to the feedback. We deleted and reduced the questionnaire options that may lead to bias and misunderstanding. We then conducted face-to-face questionnaire surveys on densely populated streets in Nanjing from April 24, 2021 to April 30, 2021, including Gulou, Xuanwu, Jianye, Qinhuai, Qixia and Yuhuatai district respectively.

6.For the purpose of drawing conclusions from the surveys, all 737 valid responses are considered ground truth and representative of the entire population of the targeted geographic region. In order to substantiate the conclusions, it will be beneficial if sensitivity analysis was performed on results. One possible approach could be providing confidence intervals on the results and regression coefficients. Another approach could be bootstrapping where a subsample of responses are selected multiple times, and the coefficients are generated. This would help quantify the variability of the results in order to determine whether the differences across various factors are within the error margins or not, in order to draw conclusions.

Reply: Thanks greatly for your useful suggestions. And we have added the confidence interval on the results and regression coefficients in the Supplementary material and revised paper (page 23, line 359).

Table 12

Stepwise regression analysis results of flood risk perception.

|                      | Model 1                  |                | Model 2                  |                | Model 3                  |               |
|----------------------|--------------------------|----------------|--------------------------|----------------|--------------------------|---------------|
| Variable      | Standardized coefficient | 95% CI         | Standardized coefficient | 95% CI         | Standardized coefficient | 95% CI        |
| Flood risk knowledge | 0.814***                 | [0.420, 0.461] | 0.827***                 | [0.427, 0.468] | =                        |               |
| Flood risk
worry  | 0.074***                 | [0.055, 0.144] | 0.067***                 | [0.046, 0.136] | 0.100**                  | [0.051,0.221] |

<sup>1 More information was shown in this website. https://www.nanjing.gov.cn/hdjl/zjdc/wsdc/dcbg/202310/t20231012 4030100.html

| Government trust               | 0.093***         | [0.033, 0.077]  | 0.094***  | [0.196,0.273] | 0.396***  | [0.196,0.273] |
|--------------------------------|------------------|-----------------|-----------|---------------|-----------|---------------|
| Flood
disaster
education | 0.060***         | [0.07, 0.254]   | 0.053***  | [0.218,0.568] | 0.146***  | [0.218,0.568] |
| Flood
experience            | -0.010*** | [-0.06, 0.033]  | 0.01      | [0.143,0.315] | 0.168***  | [0.143,0.315] |
| Gender                         | 0.057**          | [0.026, 0.13]   | Ξ         |               | Ξ         |               |
| Age                            | 0.067**          | [0.008, 0.044]  | Ξ         |               | Ξ         |               |
| District                | -0.027    | [-0.025, 0.003] | Ξ         |               | Ξ         |               |
| Education level                | 0.01             | [-0.018, 0.03]  | Ξ         |               | Ξ         |               |
| Living time                    | 0.01      | [-0.015, 0.024] | Ξ         |               | Ξ         |               |
| Health condition               | 0.056**          | [0.019, 0.077]  | Ξ         |               | Ξ         |               |
| Life style                     | 0.057**          | [0.033, 0.165]  | Ξ         |               | Ξ         |               |
| Exercise situation             | 0.038*           | [0.006, 0.099]  | Ξ         |               | Ξ         |               |
| R2                      | 0.803            |                 | 0.790     |               | 0.250     |               |
| Adjusted R2                    | 0.800            |                 | 0.788     |               | 0.246     |               |
| RMSE                    | 0.303            |                 | 0.312     |               | 0.589     |               |
| F                       | 227.27***        |                 | 549.53*** |               | 61.083*** |               |

\*\*\* P

Fig. 3. Pearson correlation analysis (The top diagonal is the regression coefficient, and the bottom diagonal is significance).

18.Section 3.4 - Briefly describe models 1, 2, and 3. From Table 12, it appears that each model used a different set of features.

Reply: Thanks for your suggestions. We used stepwise regression to reveal the influencing factors of flood risk perception. First, we selected all variables for regression analysis in model 1, and found that flood risk knowledge showed the significant and positive effect while other variables exhibited relatively lower effects. And then after removing socio-economic variables, we built the model 2 with a high goodness of fit (adjusted R²=0.788). Flood risk knowledge also maintained a higher influence (0.827) on flood risk perception. Furthermore, we excluded the variable flood risk knowledge in model 3, with a low goodness of fit (adjusted R²=0.246). But government trust, flood experience, flood disaster education and flood risk worry significantly and positively influenced risk perception, indicated by increased regression coefficients. The effect of flood experience on flood risk perception shifted from insignificant to significant. We found that although flood risk knowledge could significantly promote risk perception, it also inhibited and decreased the positive effects of other factors. And we added this brief and concise description (page 22-23, line 343-357).

Table 12 presented the results of stepwise regression analysis. The initial step involved the selection of all variables for regression analysis in Model 1. This process revealed that flood risk knowledge demonstrated a significant and positive effect, while the other variables exhibited relatively lower effects. Then after removing socioeconomic variables, this study established model 2 with a high goodness of fit (adjusted

R2=0.788). Flood risk knowledge also maintained a higher influence (0.827) on flood risk perception. Furthermore, we excluded the variable of flood risk knowledge in model 3, with a low goodness of fit (adjusted R2=0.246). But government trust, flood experience, flood disaster education and flood risk worry significantly and positively influenced risk perception, indicated by increased regression coefficients, and the effect of flood experience shifted from insignificant to significant. We found that while flood risk knowledge has the potential to significantly improve risk perception, it can also inhibit and diminish the positive impact of other contributing factors. Due to insufficient flood risk knowledge, maintaining trust in government and recalling past flooding experience were crucial for enhancing flood risk perception.

19.Sections 3.4, 3.5 - The difference in regression coefficients between various groups (e.g., males and females) appear quite small. It will be helpful to see whether these differences are in fact indicative of reality or within the error bars (such as, confidence intervals) based on the number of survey responses.

Reply: Thanks a lot for your suggestions. Although some regression coefficients are small in different groups, this study aimed to explore the differences in the influencing factors of flood risk perception and flood preparedness between different groups. It is more important that whether these different influencing factors show the significant effect or not. These small difference in regression coefficients between various groups could be accepted in this study. We had added the confidence intervals of different variables in Supplementary materials.

Table S1
Regression analysis in the gender group.

|                          | Males                    |                 | Females           |                 |
|--------------------------|--------------------------|-----------------|--------------------------|-----------------|
| Variable          | Standardized coefficient | 95% CI          | Standardized coefficient | 95% CI          |
| Flood risk knowledge     | 0.815***                 | [0.409, 0.475]  | 0.841***                 | [0.430, 0.482]  |
| Flood risk worry         | 0.087**                  | [0.047, 0.199]  | 0.043*                   | [0.001, 0.114]  |
| Government trust         | 0.105**                  | [0.027, 0.098]  | 0.090***                 | [0.026, 0.081]  |
| Flood disaster education | 0.062*                   | [0.020, 0.317]  | 0.042                    | [-0.009, 0.232] |
| Flood experience         | -0.015            | [-0.097, 0.055] | 0.027                    | [-0.022, 0.095] |
| R2                | 0.768                    |                 | 0.812                    |                 |
| Adjusted R2              | 0.764                    |                 | 0.81                     |                 |
| RMSE              | 0.340                    |                 | 0.289                    |                 |
| F                 | 209.864***               |                 | 352.248***               |                 |

\*\*\* P < 0.001, \*\* P < 0.01, \* P < 0.05

Table S2

Regression analysis in the age group.

| Variable             | Elder                    |                | Non-elder (young and middle-aged) |                |
|----------------------|--------------------------|----------------|-----------------------------------|----------------|
| variable             | Standardized coefficient | 95% CI         | Standardized coefficient          | 95% CI         |
| Flood risk knowledge | 0.828***                 | [0.374, 0.493] | 0.823***                          | [0.425, 0.469] |
| Flood risk worry     | 0.128**                  | [0.032, 0.314] | 0.059**                           | [0.032, 0.128] |

| Government trust         | 0.06         | [-0.034, 0.102] | 0.101***    | [0.037, 0.083]  |
|--------------------------|--------------|-----------------|-------------|-----------------|
| Flood disaster education | 0.042        | [-0.191, 0.411] | 0.056**     | [0.05, 0.249]   |
| Flood experience         | 0.028        | [-0.113, 0.189] | 0.007       | [-0.039, 0.059] |
| R2                | 0.78         |                 | 0.792       |                 |
| Adjusted R2              | 0.767 |                 | 0.79 |                 |
| RMSE              | 0.328        |                 | 0.310       |                 |
| F                 | 58.303***    |                 | 488.224***  |                 |

20.Fig 4 - Why are certain coefficients missing from the figure, e.g., females and flood disaster education? Same for Fig 5.

Reply: Thanks a lot for your suggestions. At first, we only listed the significant results of regression analysis in Fig. 4 and Fig. 5. Therefore, some insignificant coefficients were not presented in these figures, which we mentioned in this original manuscript (Line 330). We have redrawn Fig. 4 and Fig. 5 to avoid the missing coefficients and revealed the regression coefficients in different groups (page 25, line 386 and page 28, line 436).

Fig. 4. Regression analysis of flood risk perception.

Fig. 5. Regression analysis of flood preparedness.

21.Section 3.6 - Please describe how the influence path analysis was implemented, along with relevant references.

Reply: Thanks a lot for your suggestions. We performed a moderated mediation model in PROCESS macro program of SPSS to capture the influence path between flood risk perception and flood preparedness. The PROCESS program can effectively test the moderated mediation model and help to clarify the mediating and moderating roles of different variables. All statistical analyses were conducted at a significance level of 0.05. In this model, risk perception, flood preparedness, response intention and social-economic factors acted as independent, dependent, mediating and moderating variables respectively. And we added more detailed description and relevant references in the section of Material and methods (page 10, line 210-217).

Finally, we performed the moderated mediation model in PROCESS macro program of SPSS (Kamau-Mitchell and Lopes, 2024) to capture the influence path between flood risk perception and flood preparedness. The PROCESS program can effectively test the moderated mediation model (McMains et al., 2024) and help to clarify the mediating and moderating roles of different variables. In this model, risk perception, flood preparedness, response intention and social-economic factors acted as independent, dependent, mediating and moderating variables respectively. All statistical analyses were conducted at a significance level of 0.05.

22. Section 3.6 - New taxonomy is presented, e.g, M-1SD, without any explanation for its meaning. As a result, it was not clear how to interpret the figures and results.

Reply: Thanks a lot for your suggestions. M-1SD means that the value of a variable is one standard deviation below the mean value. We aimed to explore the moderating effect among independent, dependent, moderating variables by increasing and decreasing the level of moderating variable. In this way, we could reveal that whether the independent variable has a significant positive predictive effect on the dependent variable or not, with moderating variable being one standard deviation below (M-1SD) or above (M+1SD) its mean value. And we modified this section carefully and made it more concise and accessible (page 29-30, line

**461-470).**

This study examined the moderating and mediating effects and explored the influence path between flood risk perception and flood preparedness. Supplementary materials presented more detailed information. Risk perception, flood preparedness, response intention and social-economic factors acted as independent, dependent, mediating and moderating variables, respectively. We aimed to explore the moderating effect among independent, dependent, moderating variables by increasing and decreasing the level of moderating variable. This study could reveal whether the independent variable has a significant positive predictive effect on the dependent variable, with moderating variable being one standard deviation below (M-1SD) or above (M+1SD) its mean value.

**REVIEWER 2:**

Thank you for the opportunity to review the manuscript. The paper needs some improvement before it can be considered for publication. Here are my comments:

1. avoid the word "natural disasters". Disasters are not natural. please refer to IPCC or UNDRR reports.

Reply: Thank you for your suggestions, and we have avoided this expression in the paper.

2. The LR needs improvement. Please see Rana et al. 2020. Characterizing flood risk perception in urban communities of Pakistan. The paper reviews theories on flood risk perception. See some more references at the end for improving this section.

Reply: Thanks a lot, we would consider these references carefully and modify this section better in this paper. And this modification is shown in the section of Indtroduction (page 2-5, line 32-115).

3. Why is district taken as a socioeconomic factor? Maybe drop it from the analysis? Mean of district doesn't make sense too.

Reply: Thanks a lot, and this study focused on the urban center of Nanjing including six districts: Gulou, Xuanwu, Jianye, Qinhuai, Qixia and Yuhuatai district respectively. Based on Nanjing Statistical Yearbook, there were socioeconomic differences in these districts, and therefore, we considered it as a socioeconomic factor and mainly used it to reveal the socioeconomic differences of survey respondents in the regional distribution.

4. Figure 3. Some values of significant correlation are missing in the figure.

Reply: Thanks greatly for your valuable suggestions. We have redrawn figure 3 carefully (page 22, line 340).

Fig. 3. Pearson correlation analysis (The top diagonal is the regression coefficient, and the bottom diagonal is significance).

5. Please add a model of fitness for regression results.

Reply: Thanks a lot for your suggestions. The R-square value is a statistic that measures the goodness of fit of a regression model and indicates how well the regression model fits the

observed values. The R square value ranges from 0 to 1, and the greater the R square value, the better the regression model fits the observed value. The adjusted  $R^2$  is the correction of  $R^2$ , and the adjusted  $R^2$  takes into account the number of independent variables and the influence of sample size to avoid the problem of over-fitting. RMSE is the most commonly used evaluation model index in regression models. The closer the RMSE value is to 0, the better the model fitness. We added more descriptions about the fitness of regression analysis in Supplementary materials and this revised paper (page 23, line 359 and page 27, line 424).

Table 12 Stepwise regression analysis results of flood risk perception.

|                    | Model 1          |                 | Model 2      |                | Model 3            |                |
|--------------------|------------------|-----------------|--------------|----------------|--------------------|----------------|
| Variable    | Standardized     | 050/ CI         | Standardized | 050/ CI        | Standardized       | 050/ CI        |
|                    | coefficient      | 95% CI          | coefficient  | 95% CI         | coefficient | 95% CI         |
| Flood risk         | 0.814***         | [0.420, 0.461]  | 0.827***     | [0.427, 0.468] |                    |                |
| knowledge   | 0.014            | [0.420, 0.401]  | 0.027        | [0.427, 0.400] | Ξ                  |                |
| Flood risk         | 0.074***         | [0.055, 0.144]  | 0.067***     | [0.046, 0.136] | 0.100**            | [0.051,0.221]  |
| worry              | 0.071            | [0.022, 0.111]  | 0.007 | [0.010, 0.120] | 01100       | [0:051;0:221]  |
| Government         | 0.093***         | [0.033, 0.077]  | 0.094***     | [0.196,0.273]  | 0.396***           | [0.196,0.273]  |
| trust              |                  | -               |              | -              |                    |                |
| Flood              | 0.060***         | [0.07.0.254]    | 0.052***     | [0.210.0.560]  | 0.146***           | [0.210.0.560]  |
| disaster education | 0.060***         | [0.07, 0.254]   | 0.053***     | [0.218,0.568]  | 0.146***           | [0.218, 0.568] |
| Flood              |                  |                 |              |                |                    |                |
| experience         | -0.010*** | [-0.06, 0.033]  | 0.01  | [0.143, 0.315] | 0.168***           | [0.143,0.315]  |
| Gender             | 0.057**          | [0.026, 0.13]   | Ξ            |                | _                  |                |
| Age                | 0.067**          | [0.008, 0.044]  | _
_       |                |                    |                |
| District           | -0.027           | [-0.025, 0.003] | _
_       |                |                    |                |
| Education          |                  |                 | _            |                | _                  |                |
| level              | 0.01      | [-0.018, 0.03]  | Ξ            |                | Ξ                  |                |
| Living time        | 0.01             | [-0.015, 0.024] | Ξ            |                | Ξ                  |                |
| Health      | 0.056**          | [0.010.0.077]   |              |                |                    |                |
| condition          | 0.056**          | [0.019, 0.077]  | Ξ            |                | Ξ                  |                |
| Life style         | 0.057**          | [0.033, 0.165]  | Ξ            |                | Ξ                  |                |
| Exercise    | 0.038*           | [0.006, 0.099]  |              |                |                    |                |
| situation          | 0.038            | [0.000, 0.077]  | Ξ            |                | Ξ                  |                |
| R2          | 0.803            |                 | 0.790        |                | 0.250              |                |
| Adjusted R2        | 0.800            |                 | 0.788        |                | 0.246              |                |
| RMSE        | 0.303            |                 | 0.312        |                | 0.589              |                |
| F           | 227.27***        |                 | 549.53***    |                | 61.083***          |                |

\*\*\* P < 0.001, \*\* P < 0.01, \* P < 0.05

Table 13

Stepwise regression analysis results of flood preparedness.

| Variable | Standardized coefficients | 95% CI | p-value |  |
|-----------------|---------------------------|--------|----------------|--|
|-----------------|---------------------------|--------|----------------|--|

| Threat appraisal         | 0.213  | [0.177, 0.352]   | 0 |
|--------------------------|---------------|------------------|----------|
| Flood risk knowledge     | 0.14   | [0.040, 0.129]   | 0 |
| Flood risk worry         | 0.072         | [0.008, 0.210]   | 0.034    |
| Government trust         | 0.178  | [0.068, 0.167]   | 0 |
| Flood disaster education | 0.075         | [0.020, 0.433]   | 0.032    |
| Flood experience         | -0.078 | [-0.220, -0.016] | 0.024    |
| R2                | 0.184  |                  |          |
| Adjusted R2              | 0.177  |                  |          |
| RMSE              | 0.685  |                  |          |
| F                 | 27.439 |                  |          |

6. The manuscript is too long, maybe cut down on Mann, Kruskal-Wallis tests etc. Regression is the main thing in this paper.

Reply: Thanks a lot for your advice and correction. Mann-Whitney U and Kruskal-Wallis tests were used to compare the differences of flood risk perception and flood preparedness between variable groups, which is also important and essential in this study. And we have made this section more concise in the revised paper.

Overall, the paper is technically sound but needs a little improvement in language and flow. Minor revisions are suggested.

Minor comments:

1. Need grammar check. Especially figures and abstracts.

Reply: Thanks a lot for your advice and correction. We checked the grammar, improved the language and flow again and adjusted the figures and abstracts in this paper (page 28, line 436).

Fig. 5. Regression analysis of flood preparedness.

2. L16 Flood, not food

Reply: Thanks for this correction, and we have adjusted this word accordingly.

3. Figure 5. Flood preparedness. Check spelling.

Reply: Thanks for your correction, and we have adjusted this word accordingly (page 28, line

Fig. 5. Regression analysis of flood preparedness.

**References to consult:**

- https://doi.org/10.1016/j.ijdrr.2016.08.028
- https://doi.org/10.1016/j.ijdrr.2019.101427
- https://doi.org/10.1016/j.jenvman.2022.115309
- https://doi.org/10.1080/17477891.2023.2220947

I wish the authors well with the revision. Good luck.

Reply: Thanks a lot, we appreciated your positive assessment of our study and referred these references in Introduction section of this revised paper.

---

## Author Response (AR2)

**Details of modification**

Dear reviewers,

We would like to thank the reviewers for his/her interest in our work for their effort, constructive criticism and suggestion. We appreciate the insightful comments, as these would contribute to improving the manuscript's robustness and quality. We provide a point-by-point reply to the general and specific comments raised as follows:

**REVIEWER 1:**

As you can see the reviewers have re-reviewed your manuscript and have come to a mixed conclusion and so I have stepped in to provide a third opinion. I have reviewed the paper and have come to the conclusion that there is an interesting story in here for readers of NHESS and this is a large survey which can significantly add to the body of literature in this area. Therefore I encourage you to revise your manuscript and resubmit for further review. In doing so however please take a very careful approach to address each and every point in the review from both reviewers, as well as the additional points below..

My analysis of the paper highlights the following major issues to address:

1. this paper has become unwieldy with the presentation – there are significant numbers of tables (13) on top of 9 figures. This may have something to do with a misplaced 'supplementary materials' section - nonetheless this should be significantly revised

Reply: Thanks greatly for your valuable suggestions. We have reduced the number of forms from 13 to 11. We have placed this table about socio-economic features in respondents and descriptive statistics of each indicator and variable in the supplementary material section.

- 2. The English is poor an English language editor is required after some significant editing Reply: Thank you for your suggestions. We have polished the language style of this article and made it more understandable and concise. And this modification could be seen in the overall expression in the revised paper.
- 3. The discussion section would benefit from structure.

Reply: Thank you for your suggestions. In the Discussion section, we found that people relied more on threat appraisal to perceive risk and failed to trigger high enough coping appraisal. Insufficient risk perception led to strenuous transform into flood preparedness with unbalanced relationship. Groups with social-economic features showed different preferences to achieve risk perception and flood preparedness. And threat appraisal transformed into flood preparedness under the effect of response intention and social-economic features. Groups with high education level or bad health would more probably perceive risk and take preventive behavior. These findings could provide critical insights into intervention strategies for enhancing public flood preparedness in flood management. But this study only found the influence path in part of factors and results may not be generalized in all socio-economic characteristics. Rationality and reliability of influence path need further empirical validation in future studies. With the climate change, the adoption of different behaviors was significantly influenced by how individuals perceive and evaluate risk. When risk events were associated with adequate benefits, individuals tended to exhibit a preference for adaptive behaviors. Consequently, a thorough analysis of benefits and costs was crucial in understanding risk perception and preparedness. And this modification could be seen in the

**revised paper (page 32-36, line 548-632).**

This study found no significant gender difference in risk perception; however, females exhibited a higher level of flood preparedness, consistent with previous research (Rana et al., 2020; Rasool et al., 2022). Individuals who regularly exercised demonstrated higher risk perception, mainly because adequate physical activity enhanced their response and judgment capabilities, leading to more active cognitive functions. The elderly, particularly those aged 51-60 and above 60, showed higher risk perception but lower flood preparedness. As socially vulnerable groups, the elderly were more likely to perceive flood risk (Harlan et al., 2019), yet struggled with practical responses due to insufficient fitness and reaction capabilities. Individuals with lower education levels displayed higher risk perception, while those with higher education levels showed greater flood preparedness. People with lower educational attainment often have lower social status and are more likely to engage in hazardous occupations, motivating them to proactively perceive flood risks (Bollettino et al., 2020; Kiani et al., 2022). But highly educated individuals could access diverse information about disasters and prepare adequately for floods (Rana et al., 2020). Long living time made people become acquainted with local conditions, leading to a positive perception of flood risk. Those who experienced and worried about floods tended to perceive higher risks and made adequate preparations. Past flood experiences triggered risk perception and a greater intention to take preventive actions (Ao et al., 2020). Individuals were more likely to report higher risk perception and preparedness when floods were associated with negative emotions or memories (Rufat and Botzen, 2022).

Enough high threat appraisal could trigger coping appraisal (Schlef et al., 2018), leading to increased protection motivation and promoting mitigation measures (Kurata et al., 2022). However, our results indicate that even with high threat appraisal and moderate coping appraisal, the threat appraisal may not reach the threshold necessary to effectively trigger coping appraisal. And coping appraisal had no significant effect on flood preparedness in our study. Individuals tended to rely predominantly on threat appraisal to perceive risk, often failing to generate an adequate coping appraisal, which resulted in insufficient risk perception. Thus, risk perception struggled to translate into effective flood preparedness due to this imbalanced relationship. The influence of threat appraisal on flood preparedness was greater in groups with low risk perception compared to those with high risk perception. The transformation of low risk perception into flood preparedness could be attributed to the relatively stronger effect of threat appraisal on flood preparedness. The association between high risk perception and low flood preparedness might stem from the weaker effect of threat appraisal on flood preparedness. However, due to the significant influence of other factors, such as government trust, individuals within groups exhibiting high levels of risk perception were more likely to demonstrate greater preparedness for floods.

Various socio-economic characteristics influenced individual preferences for different methods of achieving risk perception and flood preparedness. Females exhibited higher levels of flood worry and relied more on flood knowledge to perceive risk than males, possibly due to the general cognition that women are more vulnerable and sensitive (Eryılmaz Türkkan and Hırca, 2021). Females were suggested to keep

calm, and improve risk perception through flood knowledge. The elderly depended on both flood knowledge and worry for risk perception. Although they demonstrated a greater influence of government trust on flood preparedness, lower levels of government trust could potentially hinder their efforts in flood preparedness. Individuals with low education levels preferred using flood knowledge for risk perception and were advised to enhance their trust in the government to improve flood preparedness. Those with longer residence durations relied more on flood knowledge for risk perception, while individuals with shorter living times, unfamiliar with local floods, depended more on government trust for risk perception and favored threat appraisal to achieve flood preparedness. Groups with poor health relied more on flood knowledge for flood preparedness as adequate risk knowledge could compensate for physical functional limitations. Individuals who regularly exercised showed a preference for threat appraisal in preparing for floods. Moreover, individuals with bad habits, considered psychologically fragile and sensitive, preferred flood risk worry and knowledge and government trust for risk perception.

In our study, risk perception, including both threat and coping appraisal, directly influenced flood preparedness, with response intention exhibiting a mediating effect. Socio-economic factors, especially education level and health condition, played a moderating effect between risk perception and flood preparedness. Individuals with higher education levels were better equipped to process complex information and act promptly during the time lag between action and outcome (Dootson et al., 2022). As health condition improved, there was a negative predictive effect of threat appraisal on flood preparedness. Although people reporting good health displayed confidence in their physical function, overconfidence could impede the translation of risk perception into preparedness (Bollettino et al., 2020). These groups should attach importance to timely feedback in response to floods. Among males, despite lower levels of flood preparedness, threat and coping appraisal were stronger predictors of flood preparedness. With the effect of response intention and socio-economic factors, risk perception could transform into flood preparedness, leading to differences in preventive and adaptive behaviors. Individuals with higher education levels were more likely to perceive risk and engage in preventive behavior against flooding. Conversely, groups with poorer health were more likely to perceive flood risks and adopt preventive measures.

This study revealed the influence of socio-economic factors on risk perception and flood preparedness. But we only found the influence path from a part of factors, and results may not be generalized to all socio-economic characteristics. The rationality and reliability of the identified influence paths require further empirical validation in future research. Due to climate change, the adoption of different behaviors is significantly influenced by how individuals perceive and evaluate risk (Bodoque et al., 2019). When risk events are associated with adequate benefits, individuals tend to prefer adaptive behaviors (Zhang et al., 2021b). Consequently, a comprehensive analysis of benefits and costs is crucial for understanding risk perception and preparedness.